# Zygotic pioneer factor activity of Odd-paired/Zic is necessary for late function of the *Drosophila* segmentation network

**Isabella V Soluri[1], Lauren M Zumerling[1], Omar A Payan Parra[2,3], Eleanor G Clark[2], Shelby A Blythe[1]***

[1]Department of Molecular Biosciences, Northwestern University, Evanston, United States; [2]Program in Interdisciplinary Biological Sciences, Northwestern University, Evanston, United States; [3]Department of Neurobiology, Northwestern University, Evanston, United States

**Abstract** Because chromatin determines whether information encoded in DNA is accessible to transcription factors, dynamic chromatin states in development may constrain how gene regulatory networks impart embryonic pattern. To determine the interplay between chromatin states and regulatory network function, we performed ATAC-seq on *Drosophila* embryos during the establishment of the segmentation network, comparing wild-type and mutant embryos in which all graded maternal patterning inputs are eliminated. While during the period between zygotic genome activation and gastrulation many regions maintain stable accessibility, cis-regulatory modules (CRMs) within the network undergo extensive patterning-dependent changes in accessibility. A component of the network, Odd-paired (*opa*), is necessary for pioneering accessibility of late segmentation network CRMs. *opa*-driven changes in accessibility are accompanied by equivalent changes in gene expression. Interfering with the timing of *opa* activity impacts the proper patterning of expression. These results indicate that dynamic systems for chromatin regulation directly impact the reading of embryonic patterning information.

**\*For correspondence:**
shelby.blythe@northwestern.edu

**Competing interests:** The authors declare that no competing interests exist.

## Introduction

Embryonic patterning systems direct a set of initially uncommitted pluripotent cells to differentiate into a variety of cell types and complex tissues. Over developmental time spans, regulatory networks of transcription factors drive the acquisition of unique cell fates by integrating patterning information and determining the set of genes regulated in response to developmental cues (*Levine and Davidson, 2005*). The critical nodes of these regulatory networks are cis-regulatory modules (CRMs) where transcription factors bind in order to activate or repress target gene activity. However, additional epigenetic determinants such as the organization of chromatin structure likely influence how genomic information is accessed by regulatory networks. For instance, because nucleosome positioning can hinder transcription factor-DNA interactions (*Kornberg and Lorch, 1992*; *Polach and Widom, 1995*) chromatin effectively serves as a filter either to highlight or obscure regulatory information encoded in DNA. But embryonic chromatin states themselves are dynamic (*Weintraub et al., 1981*; *McKay and Lieb, 2013*; *Xy et al., 2014*; *Blythe and Wieschaus, 2016*; *Cusanovich et al., 2018a*; *Cusanovich et al., 2018b*). The mechanisms that drive developmental progression can also trigger remodeling of chromatin accessibility patterns on both large and small scales, thereby changing over time what genetic information is available to gene regulatory systems. While in several cases we have near comprehensive understanding of both the genetic components of certain developmental networks and the critical CRMs whereby these components interact, much less is known about

how chromatin accessibility states constrain network function and how mechanisms for controlling chromatin accessibility are woven into the developmental program.

In the case of *Drosophila melanogaster*, decades of investigation into the mechanisms of development have exhaustively identified the critical patterning cues and transcription factors that drive early cell fate specification and differentiation of select developmental lineages. Patterning is initiated by four distinct maternal pathways that alone are sufficient to initiate zygotic regulatory networks that specify all the primary cell identities that arise along the major embryonic axes (*Driever and Nüsslein-Volhard, 1988*; *Anderson et al., 1985*; *Schüpbach and Wieschaus, 1986*; *Casanova and Struhl, 1989*; *Jiménez et al., 2000*; *Lehmann and Nüsslein-Volhard, 1991*; *Petkova et al., 2019*; *Hülskamp et al., 1989*). At the outset of patterning, nuclei have what can be considered a 'ground state' of chromatin structure that contains the initial set of accessible CRMs and promoters that will define the first regulatory network interactions (*Blythe and Wieschaus, 2016*). The ground state effectively provides a baseline for determining the influence of epigenetic mechanisms of gene regulation on developmental processes. The early *Drosophila* embryo therefore represents an ideal starting point to observe both how regulatory networks are constrained by chromatin states, and how these states change as a function of progression through the developmental program.

Before embryos can respond zygotically to maternal patterning cues, they must first undergo a series of 13 rapid, synchronous mitotic divisions that serve to amplify the single nucleus formed after fertilization into a set of ~6000 largely uncommitted, pluripotent cells (*Foe and Alberts, 1983*; *Farrell and O'Farrell, 2014*). These mitotic divisions occur in a state of general transcriptional quiescence that effectively prevents nuclei from responding prematurely to regulatory stimuli (*Anderson and Lengyel, 1979*; *Zalokar, 1976*; *Edgar et al., 1986*; *Edgar and Schubiger, 1986*). The shift from the initial proliferative phase to later periods of differentiation comes at a major developmental milestone termed the midblastula transition (MBT) during which the zygotic genome gradually activates, and cells become competent to respond to maternal patterning information (*Harrison and Eisen, 2015*). A major component of zygotic genome activation (ZGA) is the establishment of the chromatin ground state by a subset of transcription factors known as pioneer factors that gain access to closed or nucleosome-associated DNA and direct the establishment of short tracts of open and accessible chromatin (*Liang et al., 2008*; *Nien et al., 2011*; *Sun et al., 2015*; *Harrison et al., 2011*; *McDaniel et al., 2019*; *Schulz et al., 2015*; *Zaret and Carroll, 2011*). Because maternal pioneer factors such as Zelda are expressed uniformly in all cells (*Liang et al., 2008*), it is inferred that the initial ground state is common to all cells of the embryo at ZGA. However, maternal patterning systems can also directly influence chromatin accessibility states at ZGA (*Hannon et al., 2017*), and because their activities are by definition spatially restricted, this gives rise to heterogeneous embryonic chromatin states. Indeed, ATAC-seq measurements of chromatin accessibility in subdivided post-ZGA embryos, sampled either as posterior and anterior halves (*Haines and Eisen, 2018*), sorted cells purified from restricted positions along the anterior-posterior (AP) axis (*Bozek et al., 2019*), or as single cells (*Cusanovich et al., 2018a*) identify spatial heterogeneities in accessibility patterns that begin to emerge shortly after embryos undergo ZGA and initiate patterning. These observations raise the question of what mechanisms drive the reshaping of the chromatin landscape following ZGA.

In this study, we have investigated how chromatin accessibility states change following ZGA and to what extent these changes are dependent on the mechanisms of embryonic patterning. We find that the ZGA chromatin state must continue to change in order to support the establishment of accessible CRMs within the regulatory network that confers embryonic segmental identities. By measuring changes in chromatin accessibility over the 1-hr period between ZGA and gastrulation comparing wild-type and mutant embryos in which all graded maternal inputs to patterning are either eliminated or flattened, we define sites that display dynamic regulation of accessibility downstream of either localized pattern-dependent or global patterning-independent cues. We find that although maternal patterning systems are limited in their ability to influence directly chromatin accessibility states, distinct downstream components of zygotic gene regulatory networks make major contributions to patterning-dependent alterations of the chromatin accessibility landscape. We focus on the characterization of one such factor, Odd-paired (Opa), which we demonstrate is both necessary and sufficient to pioneer open chromatin states for a set of CRMs critical for the function of the embryonic segmentation network. These results highlight that individual components of gene regulatory

networks may operate not simply to activate or repress target gene expression, but to dictate how and when network components interact by controlling dynamic CRM accessibility.

## Results

### The ZGA chromatin state is insufficient to support embryonic segmentation

To estimate the sufficiency of the ZGA chromatin state to support early embryonic development, we scored all known enhancers in the anterior-posterior (AP) segmentation network for chromatin accessibility. During ZGA, the first zygotic components of the segmentation network are transcribed in response to maternal patterning cues (*Pritchard and Schubiger, 1996*). Over the course of nuclear cycle 14 (NC14) complex patterned gene expression from the gap, pair-rule, and segment polarity networks sequentially drive the conversion of graded maternal information into the unique segmental identities of cells across the AP axis (*Petkova et al., 2019*; *Nüsslein-Volhard and Wieschaus, 1980*; *Hülskamp and Tautz, 1991*; *Schroeder et al., 2011*; *Jaeger, 2011*). By entry into gastrulation, approximately 65 min into NC14 at 23°C, the initial activation of all three networks is complete. Because many of the CRMs required to generate the complex expression patterns of genes within the segmentation network are known, we used this system as a model to ask whether the ZGA chromatin state contains all the accessible cis-regulatory information required to complete this well-characterized developmental patterning task.

To evaluate chromatin accessibility states between ZGA and gastrulation, we collected single embryos aged either 12 or 72 min into NC14 and performed ATAC-seq. Mapped reads were assigned to peaks, which were subsequently cross-referenced against the Redfly database of previously characterized CRMs within the segmentation network (n = 111 CRMs, see *Supplementary file 1; Rivera et al., 2019*). These were then scored for accessibility either shortly after ZGA (NC14+12′) or 1 hr later at the onset of gastrulation (NC14+72′) (*Figure 1A*; see Materials and methods).

We find that CRMs within each tier of the segmentation network are not constitutively accessible and have distinct temporal chromatin accessibility profiles that correlate with the activity periods associated with these regulatory elements. All the known early gap gene CRMs are open at ZGA and either maintain or lose accessibility by the onset of gastrulation (*Figure 1A,B*). In contrast, pair-rule CRMs separate into two distinct temporal classes of chromatin accessibility. All the early, stripe-specific CRMs within the pair-rule network are open at ZGA, whereas later, seven-stripe (or 14-stripe) specific CRMs generally lack open chromatin at ZGA, and gain accessibility by the onset of gastrulation (*Figure 1A,C*). A large subset of the known segment polarity CRMs lack accessible chromatin at ZGA and undergo significant gains in accessibility by gastrulation (*Figure 1A,D*). Overall, within this network, 30% (33/111) of known CRMs undergo statistically significant changes in chromatin accessibility over time. Taken together, these results demonstrate that during the 1-hr period between ZGA and gastrulation patterns of chromatin accessibility within segmentation network CRMs are dynamic, correlating with the early or late activity of gene expression patterns within the network. We conclude from this that the ZGA chromatin state contains insufficient accessible cis-regulatory information to sustain the function of the segmentation gene network. Chromatin accessibility patterns continue to change over the 1-hr period between ZGA and gastrulation to support the later-acting components of the network, particularly the segment polarity and late pair-rule systems. This raises the possibility that the hierarchical networks that drive embryonic segmentation derive timing information from regulated chromatin accessibility. Notably, binding of pioneers (*Harrison et al., 2011*; *Rieder et al., 2017*) implicated in the establishment of the initial ZGA chromatin state is low or absent at sites that gain accessibility late (*Figure 1—figure supplement 1*), suggesting that additional, unrecognized regulatory mechanisms drive further changes in chromatin accessibility after ZGA.

### Identification of patterning-dependent and -independent changes in chromatin accessibility

The post-ZGA changes in chromatin accessibility could arise either uniformly within all cells of the embryo or could stem from the localized effects of developmental patterning systems. Previous investigations of chromatin accessibility states in post-ZGA embryos have demonstrated that shortly

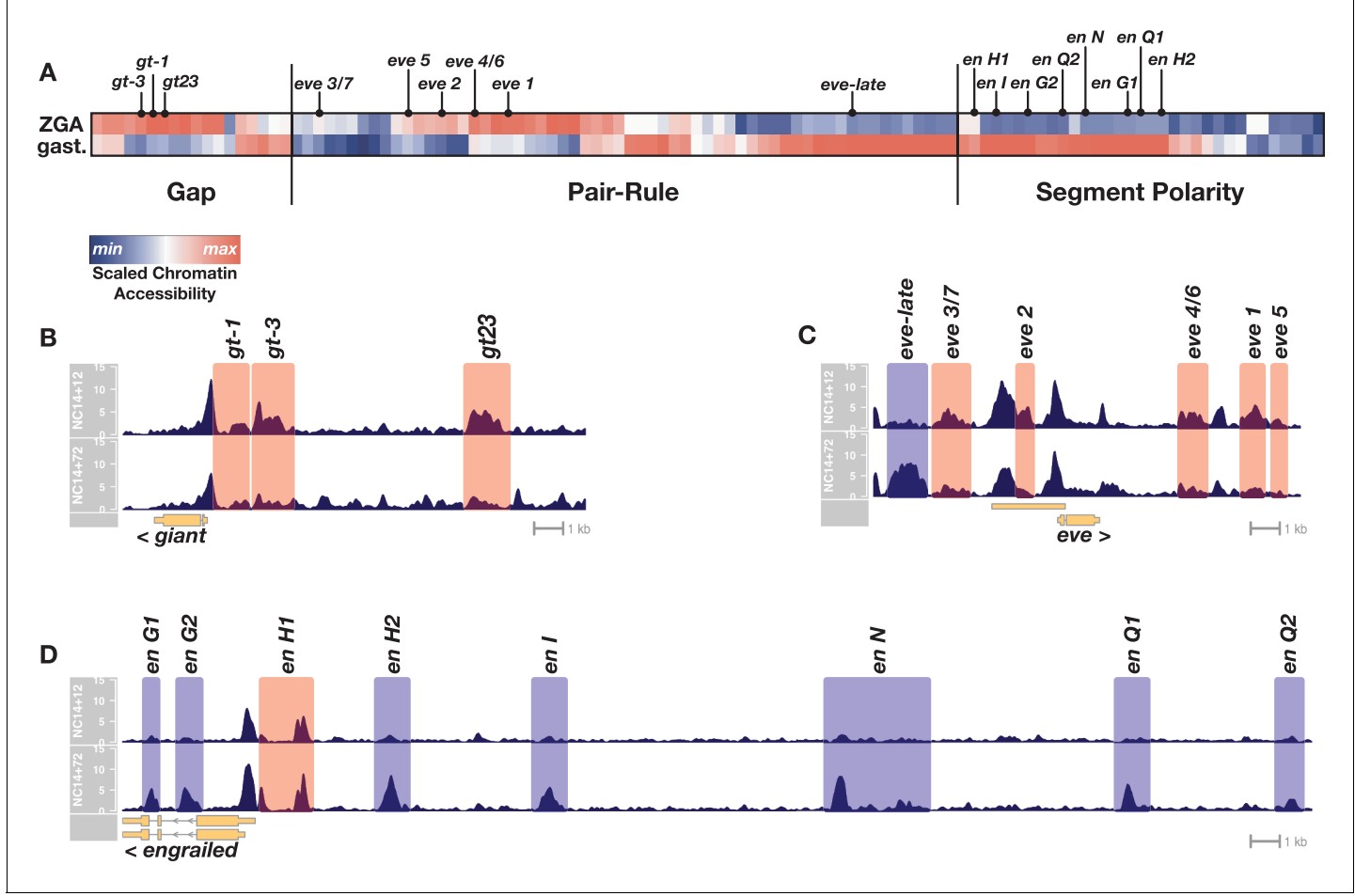

**Figure 1.** Chromatin accessibility at segmentation network enhancers is dynamic over the 1-hr period between ZGA and gastrulation. (**A**) Heatmap showing the scaled chromatin accessibility between ZGA (top row, NC14 + 12') and gastrulation (bottom row, NC14 + 72') over the complete set of known enhancers within the gap, pair-rule, and segment polarity gene regulatory networks. Tiers of the segmentation network are indicated as well as selected enhancers from the specific examples depicted in panels B-D. (**B–D**) chromatin accessibility at a representative gap (giant), pair-rule (eve), and segment polarity (engrailed) locus between ZGA (NC14+12, top) and gastrulation (NC14+72, bottom). Enhancers that are significantly more open by ZGA are highlighted in red, those that open late, by the onset of gastrulation are highlighted in blue. Loci are drawn to the same genomic (x-axis) scale, and ATAC accessibility is shown at the same y-axis scale (0–15 CPM) across all plots. See also *Figure 1—figure supplement 1*.

The online version of this article includes the following figure supplement(s) for figure 1:

**Figure supplement 1.** Binding of known early pioneers and Opa to genomic regions of three representative segmentation network factors.

after ZGA, chromatin accessibility is largely homogeneous across most cells of the embryo (*Hannon et al., 2017*; *Haines and Eisen, 2018*), but as development proceeds, spatially restricted, lineage-specific patterns of accessibility emerge as cell fates are specified and spatially restricted gene expression programs mature (*Cusanovich et al., 2018a*; *Bozek et al., 2019*). One major exception to ZGA chromatin state homogeneity is a small cohort of enhancers with anteriorly restricted accessibility whose open state depends on the activity of the maternal patterning factor Bicoid (Bcd) (*Cusanovich et al., 2018a*; *Hannon et al., 2017*; *Haines and Eisen, 2018*). Besides Bcd, it is unclear whether any other maternal patterning systems or downstream zygotic targets directly impact the chromatin accessibility state of the early embryo. We therefore measured the effect of eliminating all graded maternal inputs to embryonic development on the changes in chromatin accessibility we observe between ZGA and gastrulation.

By eliminating or flattening all graded maternal inputs to development, we force all cells of the embryo to develop along a single restricted trajectory. In situ hybridization of wild-type NC14 embryos for representative markers of anterior-posterior (AP) and dorsal-ventral (DV) specification

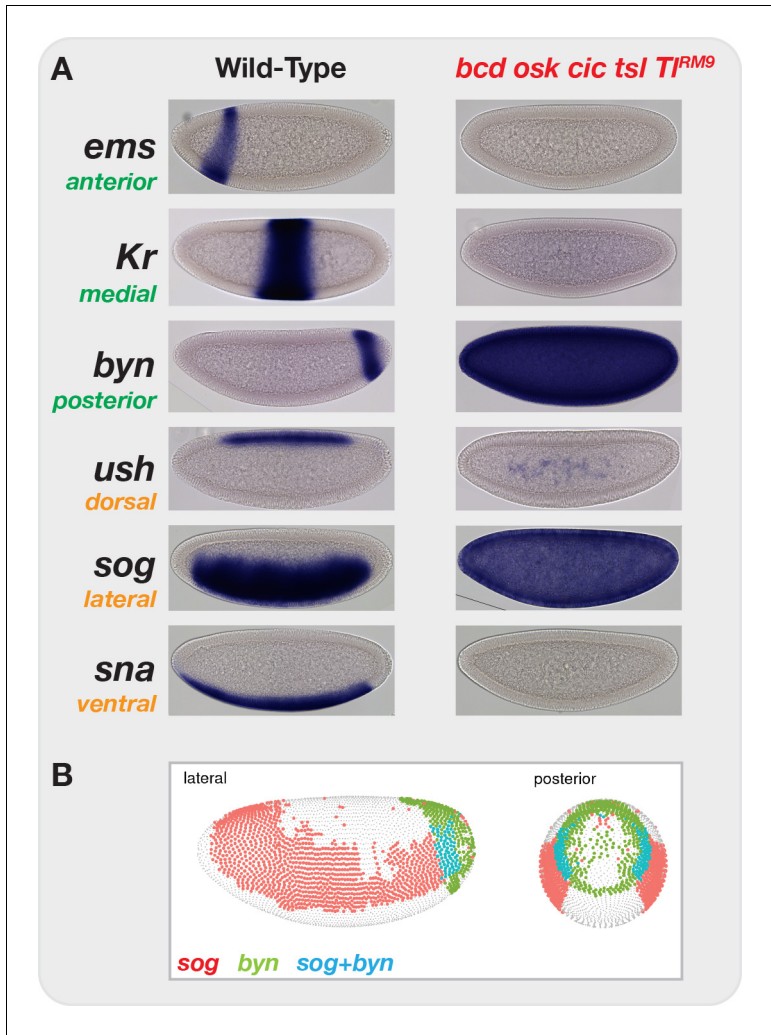

**Figure 2.** Elimination of graded maternal cues drives development along single uniform lineages. (**A**) In situ hybridization for markers of anterior-posterior (*ems*, *Kr*, *byn*) and dorsal-ventral (*ush*, *sog*, *sna*) marker genes in both wild-type and *bcd osk cic tsl Tl^{RM9}* mutant embryos. Elimination of graded positional information converts all cells in the blastoderm embryo to posterior-lateral cell types (*sog+ byn+*). (**B**) Image from the DVEX virtual expression explorer shows the subset of cells co-expressing both *sog* and *byn*. Note, for presentation, certain images were rotated from their original positions and missing background pixels were filled in. No pixels corresponding to the embryo were altered.

reveals distinct, spatially restricted gene expression patterns along both axes (*Figure 2A*, left column). Four maternal systems provide the information sufficient to distinguish these positional identities along the AP and DV axes. The terminal extrema of the embryo are distinguished by localized Torso receptor tyrosine kinase antagonism of the medial determinant Capicua (*Casanova and Struhl, 1989*; *Jiménez et al., 2000*). Anterior fates are specified by the Bcd transcription factor gradient (*Driever and Nüsslein-Volhard, 1988*; *Struhl et al., 1989*). Posterior identity is conferred by clearance of maternal Hunchback transcriptional repressor by graded expression of the Nanos translational repressor (*Lehmann and Nüsslein-Volhard, 1991*; *Hülskamp et al., 1989*; *Ephrussi et al., 1991*). DV positions are specified by a gradient of the transcription factor Dorsal that functions downstream of a ventral-to-dorsal gradient of activated Toll receptor (*Roth et al., 1989*). We generated embryos from mothers with null mutations in all terminal and AP systems (*bcd*, *osk*, *cic*, *tsl*) as well as a 'lateralizing' allele of *Toll* (*Tl^{RM9}*) that yields uniform, mid-level Dorsal activity across the entire DV axis (*Roth et al., 1989*) (see Discussion). All cells in these quintuple mutant embryos are

positive for markers of lateral (*sog*) and posterior-terminal (*byn*) positional markers and do not express markers from other regions of the embryo (*Figure 2A*, right column). By reference to a previously reported single-cell RNA-seq dataset (*Karaiskos et al., 2017*), dual *sog+ byn+* cells constitute a small fraction of posterior-lateral cells at this stage of development (*Figure 2B*) that are all fated to become posterior endoderm. We predicted that, compared with wild-type embryos, *bcd osk cic tsl Tl^{RM9}* (hereafter, 'mutant') embryos would enrich for posterior endodermal chromatin states at the expense of all others, allowing for the unambiguous determination of sites that undergo patterning-dependent versus -independent changes in accessibility.

We therefore performed ATAC-seq comparing single wild-type or mutant embryos precisely staged at 12 and 72 min into NC14. Over this early period, we expect that embryos from these two genotypes develop with similar rates and therefore remain comparable until at least the time when wild-type embryos gastrulate. As expected, comparison of differentially accessible regions between stage and genotype identifies regulatory elements with spatially restricted expression patterns. For example, at the *wingless* (*wg*) locus, two closely apposed regions (*wg* −2.5 and −1) show differential behavior between genotypes. The *wg* −2.5 region undergoes a modest increase in accessibility in wild-type embryos but not in mutants (*Figure 3A*, green shading). In contrast, the neighboring *wg* −1 region shows greatly increased accessibility in mutant samples (*Figure 3A*, cyan shading). These two regions together constitute a single previously identified CRM (*wg WLZ4L*) controlling early *wg* expression in both a segment-polarity multi-stripe pattern as well as a single posterior endodermal stripe (*Lessing and Nusse, 1998*). On the basis of our ATAC data, we individually cloned the *wg* −2.5 and −1 regions into reporter constructs and compared expression between wild-type and mutant embryos. *wg* −2.5 becomes active just prior to gastrulation in wild-type but not in equivalently staged mutant embryos, and exclusively drives multi-stripe expression within the segmental primordium (*Figure 3B* and data not shown). In constrast, *wg-1* is active earlier in NC14 and exclusively drives expression of the endodermal stripe pattern. As expected, whereas *wg* −1 expression is restricted to a single posterior endodermal domain in wild-type embryos, all cells in a mutant embryo have strong *wg* −1 expression (*Figure 3B*). We therefore performed differential enrichment analysis using DESeq2 (*Love et al., 2014*) to identify the complete set of regions with patterning-dependent or -independent changes in chromatin accessibility.

We identify significant sources of both pattern-independent as well as patterning-dependent changes in chromatin accessibility over the period between ZGA and gastrulation. In general, a greater fraction of the sites with dynamic chromatin accessibility undergo patterning-independent changes. This was quantified in two ways, and source code for the following analysis has also been provided (*Source code 1*). First, we performed principal component analysis (PCA) on the complete set of differentially accessible regions (i.e. all regions with both a ± 2 fold change and an adjusted p-value>0.01 either between timepoints or between genotypes as determined by DESeq2). The greatest source of variance is patterning-independent, with the first principal component separating samples according to developmental time (PC1: 66% variance, *Figure 3C*). The second principal component separates samples according to genotype and therefore resolves patterning-dependent variance (PC2: 21% variance, *Figure 3C*). There is less of a patterning-dependent difference between NC14+12′ samples compared with the +72′ timepoint, supporting the conclusion that cells initiate the zygotic phase of development with a large degree of chromatin state homogeneity and that additional changes emerge over the period leading up to gastrulation from patterning-dependent and -independent sources.

To relate the observed changes to discrete developmental processes, we returned to the set of known segmentation network CRMs and plotted scaled chromatin accessibility over time between wild-type and mutant samples. Within all three tiers of the segmentation network, we find evidence for extensive patterning-dependent chromatin accessibility at both early and late timepoints (*Figure 3D*). As previously shown, the gap gene network receives extensive patterning-dependent chromatin accessibility cues from a pioneer activity of Bcd (7/18 CRMs, 39%) (*Hannon et al., 2017*). Early pair-rule CRMs receive both patterning dependent and independent inputs; however, the majority of late pair-rule CRMs gain accessibility in a patterning-dependent manner (25/30 pair-rule CRMs with late accessibility, 83%). Segment polarity CRMs (e.g., *wg* −2.5 and *wg* −1, *Figure 3A*) likewise have extensive patterning-dependent accessibility states (16/33, 49%). Overall, 43% (48/111) of these segmentation network CRMs receive accessibility cues either directly or indirectly from maternal patterning systems. Therefore, these results indicate that although overall changes in

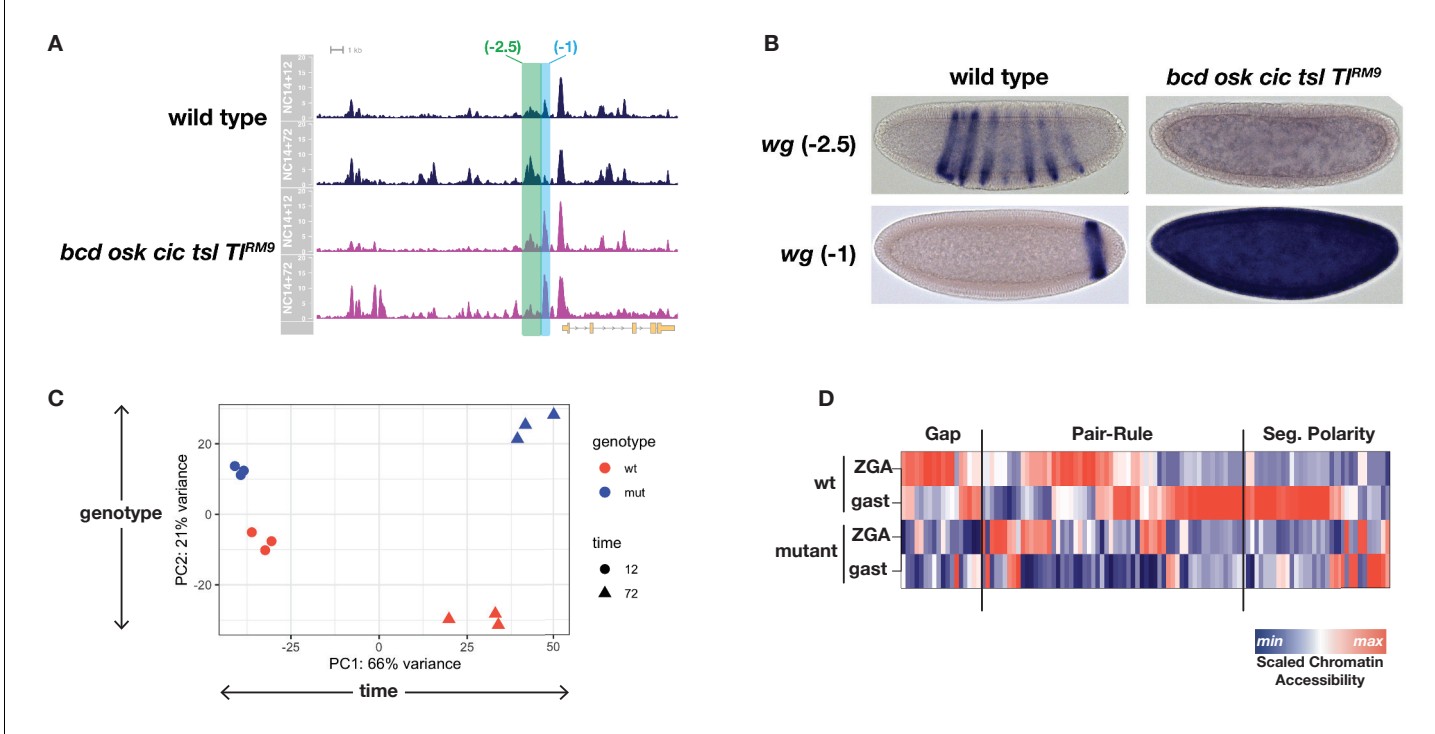

**Figure 3.** ATAC-seq on wild-type and uniform-lineage embryos resolves time and patterning-dependent changes in chromatin accessibility. (**A**) ATAC-seq coverage over the extended *wingless* locus. Highlighted regions show closely apposed CRMs with differential responses to patterning inputs. The scale for the y-axis is 0–20 CPM for all plots. (**B**) Reporters for the CRMs highlighted in panel A demonstrate separable regulatory inputs into the *wg* locus, and differential regulation in uniform-lineage embryos. (**C**) Principal component analysis of dynamically accessible regions demonstrates the relative contribution of uniform (time) and patterning (genotype) drivers of chromatin accessibility. (**D**) Heatmap showing the scaled chromatin accessibility between ZGA and gastrulation for wild-type and mutant embryos over the complete set of known enhancers within the gap, pair-rule, and segment polarity gene regulatory networks. Colorbar indicates the scaled degree of chromatin accessibility plotted in each column. See also *Figure 3—figure supplements 1–4*.

The online version of this article includes the following figure supplement(s) for figure 3:

**Figure supplement 1.** Differential accessibility between wild-type and mutant samples at NC14+12′: This volcano plot shows the effect size for comparisons between genotypes at ZGA (NC14+12′).

**Figure supplement 2.** Differential accessibility between wild-type and mutant samples at NC14+72′: This volcano plot shows the effect size for comparisons between genotypes at gastrulation (NC14+72′).

**Figure supplement 3.** Differential accessibility between ZGA and gastrulation in wild-type embryos: This volcano plot shows the effect size for comparisons between timepoints (NC14 + 12′ and + 72′) in wild-type embryos.

**Figure supplement 4.** Differential accessibility between ZGA and gastrulation in mutant embryos: This volcano plot shows the effect size for comparisons between timepoints (NC14 + 12′ and + 72′) in mutant embryos.

accessibility tend to occur independently of embryonic patterning, the networks dedicated specifically to embryonic patterning display a disproportionate reliance on patterning systems for determination of their chromatin accessibility states.

Next, we quantified the types of changes in chromatin accessibility that we observed in our analysis. Overall, there are 26,328 peaks in the ATAC dataset. Similar to the PCA analysis, we find fewer patterning-dependent differences at ZGA than at gastrulation (408 versus 1871, *Figure 3—figure supplements 1* and *2*). In contrast, a greater number of time-dependent differences are observed for both genotypes (5190 for wild-type, 8655 for mutant, *Figure 3—figure supplements 3* and *4*). We note that these numbers represent above-threshold statistical significance for tests on only one of two critical parameters in this experiment, either time or genotype.

To comprehensively classify the types of changes that any single region undergoes over time and relative to patterning inputs, we took a clustering-based approach to identify groups of similarly-behaved regions and then used the output from paired DESeq2 tests to assign regions to each identified category (see Materials and methods). By this approach, we identify a total of 2917 dynamic

regions (11% of the total peaks list, n = 26328) that classify into one of ten distinct dynamic categories with respect to time and patterning-dependence (*Figure 4*, and *Figure 4—figure supplements 1–10*, please see *Supplementary file 2* for a complete ATAC peaks list). Overall, roughly similar numbers of sites gain and lose accessibility over time with 46% of sites that are open early losing accessibility over time (1351/2917, 'late repression' classes, *Figure 4*, groups 1 through 5) and 50% sites gaining accessibility by the onset of gastrulation (1446/2917, 'late accessibility' classes, *Figure 4*, groups 8 through 10). In total we identify 56% (1635/2917) strictly pattern-independent regions that either gain (25% (725/2917)) or lose (31% (910/2917)) accessibility uniformly between ZGA and gastrulation (*Figure 4*, groups 1 and 8, see *slam* detail, and *Figure 4—figure supplements 1* and *8*). The remaining 44% of regions (1282/2917) receive inputs from patterning systems. 11% (307/2917) additional regions gain accessibility uniformly by ZGA but undergo patterning-dependent losses in accessibility by gastrulation (*Figure 4*, groups 2 and 3, and *Figure 4—figure supplements 2* and *3*). In this case, we cannot distinguish whether the patterning-dependent behavior is repressive or instead represents spatially restricted maintenance of accessibility against a uniform repressor. The remaining classes all reflect pattern-dependent behaviors. 9% of regions (254/2917) show pattern-dependent early accessibility (*Figure 4*, groups 4 through 7). This class is split either into regions that lose accessibility by gastrulation (n = 134, *Figure 4* group 4, see *gt-10* detail, and *Figure 4—figure supplements 4* and *5*) or show constitutive patterning-dependent accessibility through gastrulation (n = 120, *Figure 4* groups 6 and 7, and *Figure 4—figure supplements 6* and *7*). Finally, 25% regions (721/2917) demonstrate late patterning-dependent gains in accessibility (*Figure 4* groups 9 and 10, see *slp1 '5'* detail, and *Figure 4—figure supplements 9* and *10*). Patterning-dependent regulation of chromatin accessibility therefore increases by a factor of 4-fold over the course of NC14. While patterning-dependent accessibility at ZGA is limited to 254 regions, over the course of NC14, patterning systems have a significant impact on the embryonic chromatin landscape driving late gains in accessibility at 721 sites and late reductions in accessibility (or maintenance, see above) at an additional 307 sites for a total of 1028 late patterning-dependent regions.

These sites undergoing dynamic chromatin regulation are largely enriched for intergenic and intronic regions and are depleted for promoter regions (*Figure 4—figure supplement 11*). Compared with the 42 min of development that precede large-scale ZGA (NC11 through NC13) where thousands of promoters display dynamic acquisition of chromatin accessibility (*Blythe and Wieschaus, 2016*), the period between ZGA and gastrulation shows largely invariant promoter chromatin accessibility. Dynamic chromatin regulation during this time is focused on putative intergenic and intronic regulatory elements. However, in addition, nearly every group that displays early chromatin accessibility and gastrula-stage reductions in accessibility (*Figure 4*, groups 1–4) show an enrichment for exonic regions. While this raises the possibility that groups 1 through 4 additionally capture exonic regions with differential expression over this period, intersecting this set of exons with a published RNA-seq timecourse (*Lott et al., 2011*) reveals that few of these exons are associated with contemporaneous zygotic transcriptional activity. Across all groups, an average of 8 ± 5% of associated exons are zygotically expressed during NC14 (*Figure 4—figure supplement 11*).

To investigate the functional implications of enriched intergenic and intronic regions, we cross-referenced the collection of functionally-validated enhancer elements from the Vienna Tiles collection (*Kvon et al., 2014*) to address whether these groupings enriched for CRMs with related expression patterns. The expression patterns of individual Vienna Tiles have been extensively annotated (http://enhancers.starklab.org) and we tested for enrichment of controlled-vocabulary (CV) terms describing the expression domains of enhancers overlapping the different ATAC peak class groupings. Overall, enhancers associated with regions that undergo patterning-independent regulation of chromatin accessibility (groups 1 and 8) nevertheless demonstrate spatially restricted patterns of expression. Early, patterning-independent regions (group 1) are enriched for expression domains that span the AP and DV axes (*Figure 4—figure supplement 1D*). Late patterning-independent regions (group 8) likewise demonstrate restricted patterns of expression with enrichment for anterior ectodermal and endodermal domains (*Figure 4—figure supplement 8D*). These observations are consistent with an interpretation that patterning-independent changes in chromatin accessibility underlie not the spatial extent but rather the timing of expression for associated enhancers. One caveat to this conclusion is that the Vienna Tiles collection is itself enriched for regions likely to have spatially restricted, as opposed to ubiquitous, expression patterns and may under-represent the potential enrichment of more broadly active CRMs.

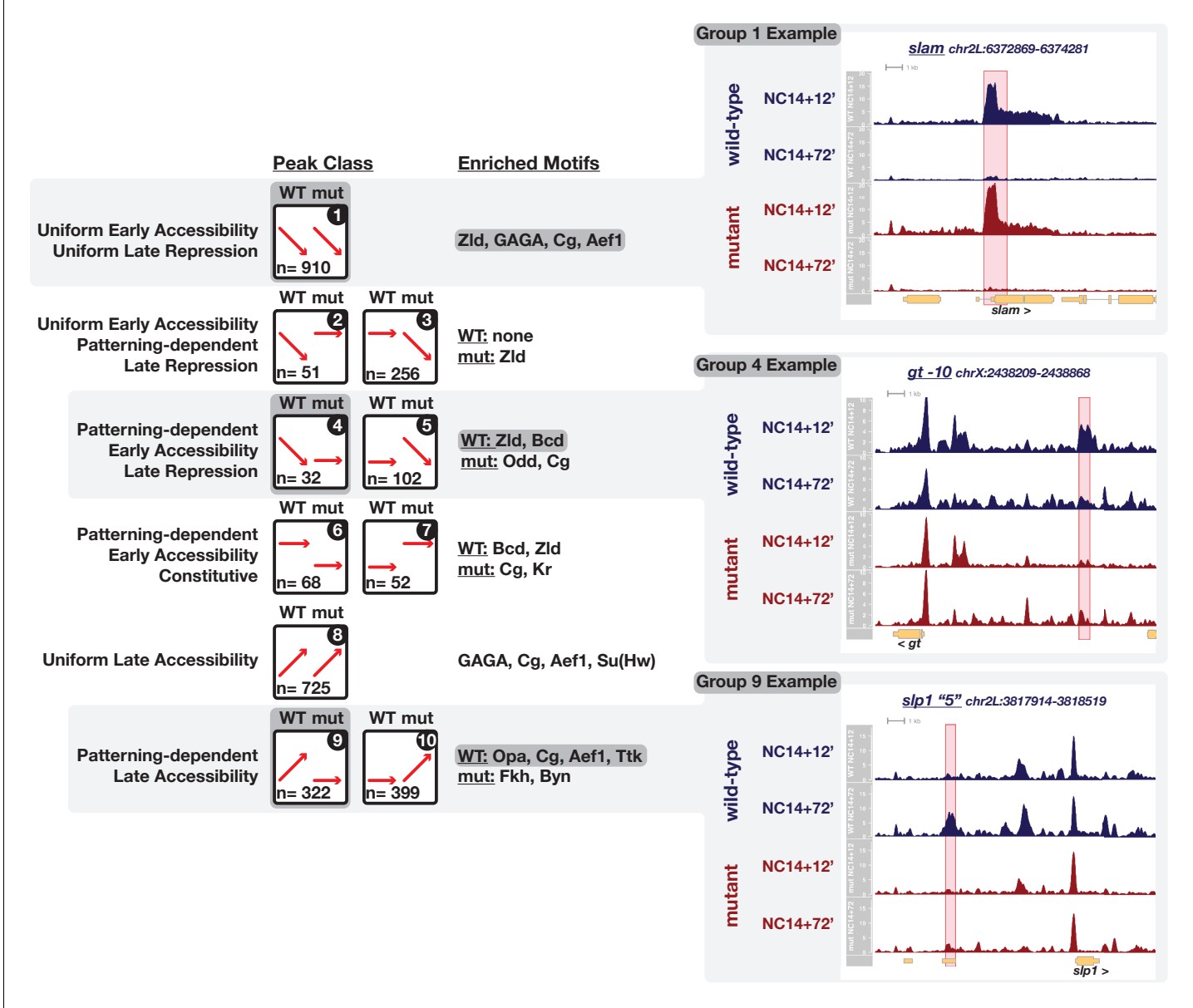

**Figure 4.** Ten classes of peaks with dynamic chromatin accessibility between ZGA and gastrulation. Ten peak classes are symbolized by cartoons in the left column. Each grouping of peaks has been numbered for reference in the text. Supplements to this figure (*Figure 4—figure supplements 1–10*) detail the features of each grouping. In the cartoons, red arrows signify the general behavior of peaks within a class between ZGA and gastrulation for wild-type and uniform-lineage mutant embryos. The number of regions per class is indicated in each cartoon. The center column lists enriched motifs as determined by MEME analysis against a set of $1 \times 10^4$ non-dynamic control regions. Examples of three highlighted groups are shown on right. The cartoon and enriched motifs associated with the example groups are additionally highlighted with dark grey boxes. The right column shows example ATAC-seq coverage plots for the three highlighted groups (1, 4 and 9, see Figure Supplements for examples of all groups). The specific peak region within the class is highlighted in red, and we note that additional non-highlighted dynamic regions may be present. See also *Figure 4—figure supplements 1–11*.

The online version of this article includes the following figure supplement(s) for figure 4:

**Figure supplement 1.** Uniform early accessibility with uniform late repression.

**Figure supplement 2.** Uniform early accessibility with late wild-type repression.

**Figure supplement 3.** Uniform early accessibility with late mutant repression.

**Figure supplement 4.** Wild-type early accessibility with late repression.

**Figure supplement 5.** Mutant early accessibility with late repression.

**Figure supplement 6.** Wild-type early accessibility 'constitutive'.

**Figure supplement 7.** Mutant early accessibility, 'constitutive'.

*Figure 4 continued on next page*

*Figure 4 continued*

**Figure supplement 8.** Uniform late accessibility.
**Figure supplement 9.** Wild-type late accessibility.
**Figure supplement 10.** Mutant late accessibility.
**Figure supplement 11.** Enrichment of generic genomic features within differentially accessible groups.

Enhancers associated with patterning-dependent changes in chromatin accessibility demonstrate a greater degree of spatial restriction in their expression patterns. As observed for the *wg* −1 enhancer (*Figure 2A,B*), increased or persistent accessibility in mutant embryos correlates with expression domains largely limited to posterior and endodermal expression patterns (*Figure 4—figure supplements 2D* and *10D & E*). Group 3, in which accessibility is lost in mutant but not wild-type embryos, demonstrates a corresponding lack of enhancer activity within the posterior endodermal compartment (e.g. VT39546 and VT7922, *Figure 4—figure supplement 3D*). Conversely, increased or persistent accessibility in wild-type embryos corresponds to expression domains that largely exclude the posterior endodermal compartment. Groups 4 and 6, which are enriched for Bcd binding motifs, are likewise enriched for enhancers with anteriorly restricted expression domains, and any posterior expression domains exclude posterior endodermal precursors (*Figure 4—figure supplements 4D* and *6D*). Finally, the late, wild-type patterning-dependent group (group 9) demonstrates enrichment for enhancers with AP or pair-rule stripes (e.g. VT15159 and VT26324). A subset of enhancers in group nine also demonstrate AP stripes with graded or discontinuous modulation along the DV axis (e.g. VT7841 and VT40612, *Figure 4—figure supplement 9D*), thus raising the possibility that these enhancers receive regulatory inputs from both the AP and DV patterning systems. Our groupings therefore primarily distinguish between enhancer subclasses that have activities associated with, or distinct from, posterior endodermal precursors and may help to define a regulatory network for early endodermal development.

Regions with patterning-dependent accessibility regulation are enriched for DNA sequence motifs associated with patterning factors. To gain insight about what regulatory factors could be driving the observed dynamic chromatin accessibility behaviors, we performed sequence motif enrichment analysis within each dynamic class using the MEME suite (*Bailey et al., 2015*). Regions with strictly uniform regulation of accessibility are enriched for motifs associated with ubiquitously expressed maternally supplied regulators, including binding sites for Zld (*Liang et al., 2008*) and GAGA/CLAMP (*Rieder et al., 2017*; *Tsukiyama et al., 1994*; *Farkas et al., 1994*; *Figure 4*, groups 1 and 8). In contrast, patterning-dependent sites are enriched for regulators with patterned, spatially restricted expression such as Bcd, Odd-paired (Opa), and Forkhead (Fkh, *Figure 4*, groups 4, 6, 9, and 10). In addition to these, we frequently observe across all categories enrichment of motifs for three maternally supplied factors with expected repressive activity, Tramtrack (Ttk), Adult Enhancer Factor 1 (Aef1), and Combgap (Cg) (*Figure 4*, groups 1, 5, 7, and 9). While Ttk has long been hypothesized to play a broad repressive role over the maternal-to-zygotic transition (*Pritchard and Schubiger, 1996*) as well as in regulation of embryonic patterning (*Harrison and Travers, 1990*; *Brown and Wu, 1993*; *Read et al., 1992*; *Wheeler et al., 2002*), much less is known about potential early embryonic roles of Aef1 and Cg (see Discussion). We also note that although motifs for Cg are not included in the available DNA binding motif databases used to compile these results, we include Cg here on the basis of previous identification of Cg binding to a $(CA)_n$ motif (*Ray et al., 2016*), maternal expression of Cg, and frequent recovery of an orphan motif $(CA)_n$ in our analysis.

The recovery of Bcd and Fkh motifs in our dataset suggests that enrichment analysis could identify potential patterning-dependent pioneer activities responsible for driving differential accessibility states. We note here that recovery of motifs in our analysis is similar to those recovered in another recent report (*Nevil et al., 2019*), which measured changes in accessibility between early and late NC14 samples but did not distinguish between patterning-dependent and independent events. Bcd has been demonstrated to pioneer accessibility at a subset of its targets (*Hannon et al., 2017*), and Bcd-motif enrichment in this analysis correlates with the set of previously identified bcd-dependent regions (e.g. *gt* −10). Fkh is the *Drosophila* homolog of a well-characterized pioneer factor FoxA1/2 that operates in early mammalian endodermal development (*Iwafuchi-Doi et al., 2016*; *Cirillo et al., 2002*; *Ang et al., 1993*). Fkh is expressed zygotically late in NC14 within the posterior endodermal

precursors we enrich with mutant embryos, and Fkh motif enrichment is observed specifically within the set of regions that have enhanced late accessibility in mutant embryos (*Figure 4*, group 10). Therefore, it is possible that like its mammalian counterpart, Fkh may pioneer accessibility of distinct endodermal CRMs in early *Drosophila* development.

Within the set of regions with patterning-dependent late accessibility, we also enrich for a zygotic pair-rule transcription factor, Opa (*Figure 4*, group 9, and *Figure 4—figure supplement 9*). Opa is a C2H2 zinc-finger transcription factor that is the *Drosophila* homolog of Zn-Finger of the Cerebellum (Zic) proteins, which have been implicated in a broad range of developmental functions ranging from maintenance of stem cell pluripotency, neural crest specification, and neural development (*Houtmeyers et al., 2013*; *Luo et al., 2015*; *Groves and LaBonne, 2014*; *Vásquez-Doorman and Petersen, 2014*; *Hursh and Stultz, 2018*). Broad roles for Opa in *Drosophila* development are also likely. In addition to an early requirement of Opa for segmental patterning, Opa is necessary for mid-gut morphogenesis (*Cimbora and Sakonju, 1995*), imaginal disc and adult head development (*Lee et al., 2007*), and has recently been shown to play a critical role in regulating the temporal identity of neural progenitors (*Abdusselamoglu et al., 2019*). Although neither Opa nor its homologs have been shown previously to pioneer chromatin accessibility, mouse Zic2 has been demonstrated to bind both active and poised enhancers in embryonic stem cells prior to Oct4 and to play a critical role in maintenance of stem cell pluripotency (*Luo et al., 2015*), and in certain species of moth flies Opa has been adopted as the maternal anterior determinant (*Yoon et al., 2019*), which in *Drosophila* requires pioneer activity from Bcd (*Hannon et al., 2017*). This, coupled with the observed enrichment of Opa motifs in our analysis, raised the possibility that Opa could contribute to patterning-dependent chromatin regulation after ZGA.

## Opa pioneers chromatin accessibility of late patterning-dependent regions

Originally identified in a genetic screen for regulators of embryonic segmentation (*Jürgens et al., 1984*), *opa* functions as a pair-rule gene to pattern alternating segmental identities. However, *opa* differs from the other seven pair-rule genes in several ways. First, *opa* is not expressed in a characteristic early seven-stripe pattern typical of pair-rule genes but is instead expressed uniformly over the entire embryonic segmental primordium (*Benedyk et al., 1994*; *Figure 5A*). Second, compared with other pair-rule genes, *opa* initiates expression significantly later in development and has been proposed to function as a temporal switch within the pair-rule network, facilitating the transition from early to late phases of network operation (*Clark and Akam, 2016*). During this transition, new regulatory interactions between network components are observed (e.g. emergence of repressive interactions at gastrulation between *paired* and *odd-skipped* within cells where both factors are co-expressed at earlier timepoints) (*Clark and Akam, 2016*), and the mechanism whereby Opa mediates these new network interactions is unclear. Because we observed enrichment of Opa binding motifs in the set of regions that gain patterning-dependent accessibility late in NC14, and because many pair-rule and segment polarity CRMs also display late patterning-dependent accessibility, we tested the hypothesis that Opa functions as a pioneer factor to confer changes in chromatin structure downstream of maternal patterning cues.

By ChIP-seq for RNA Pol2 (*Blythe and Wieschaus, 2015*), whereas typical pair-rule genes (e.g. *ftz*) show robust Pol2 association as early as NC12, little if any Pol2 is detected at the *opa* locus until NC13, when a small peak of Pol2 forms at the promoter. Productive elongation of *opa* becomes apparent at the beginning of NC14 (*Figure 5—figure supplement 1*). To measure *opa* expression dynamics, we used CRISPR to insert an anti-GFP llama tag (*Bothma et al., 2018*; *Kirchhofer et al., 2010*) to the 3' end of the *opa* coding sequence and live-imaged Opa-llama protein expression in embryos co-expressing free maternal EGFP. Engineered *opa-llama* flies are homozygous viable and show no detectable adverse effects from manipulation of the *opa* locus. Consistent with RNA Pol2 measurements, no Opa protein is detected prior to NC14 (data not shown). Llama-tagged Opa first becomes detectable above background by live imaging at 35 min into NC14 reaching an apparent steady-state expression level at 60 min, shortly before gastrulation (*Figure 5B* and *Video 1*). These measurements indicate that Opa expression is consistent with an exclusively late NC14 role in regulating gene expression over nearly all cells of the embryonic segmental primordium.

By measuring chromatin accessibility in single $opa^{+/+}$, $opa^{-/+}$, and $opa^{-/-}$ embryos at NC14 + 72', we find that Opa is necessary to pioneer chromatin accessibility at a subset of its direct genomic

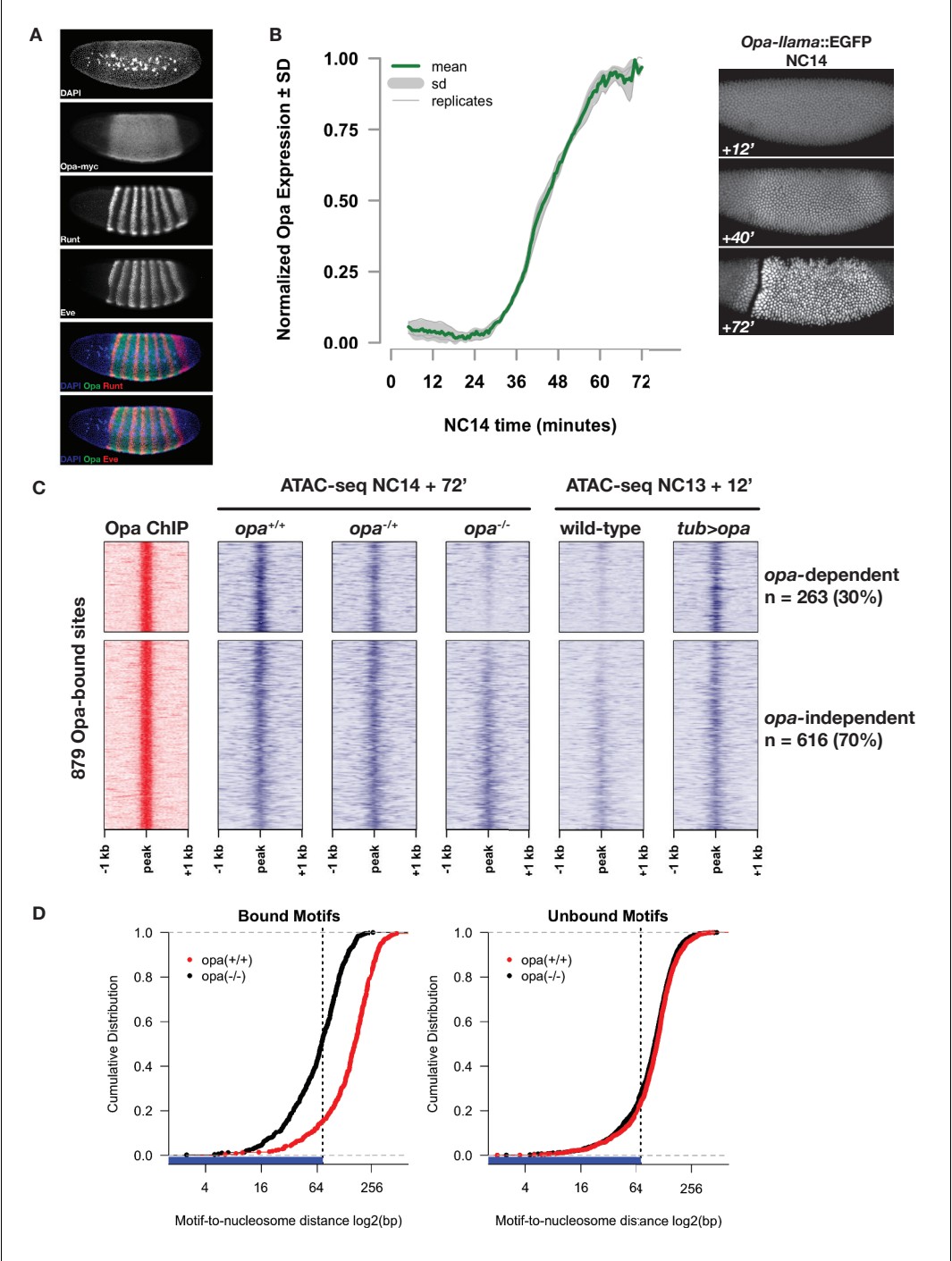

**Figure 5.** Opa is necessary and sufficient to pioneer accessible chromatin. (**A**) Immunostaining for myc-tagged Opa expression relative to expression of pair-rule genes Runt and Eve in a late NC14 embryo. (**B**) Opa expression dynamics were measured using a llama-tagged *opa* allele. Opa expression initiates midway through NC14 and reaches steady levels by entry into gastrulation at 65 min. Images show representative expression of *opa-llama*::EGFP at the indicated timepoints. (**C**) Heatmaps showing scaled ATAC-seq accessibility measurements (blue) over a set of high-confidence Opa binding sites, as determined by ChIP-seq for Opa-myc (red). Two experiments are shown, for loss of function at NC14 + 72', and gain of function at NC13 +12. Loss of blue signal indicates a reduction in accessibility. (**D**) Cumulative distribution of measured of distances between bound (left) or unbound (right) Opa motifs and modeled nucleosome dyad positions, in the absence (black) or presence (red) of Opa. X-axis is log2 scaled, and the expected coverage of a nucleosome is depicted by the blue rug and vertical dotted line. See also *Figure 5—figure supplements 1–4*.

The online version of this article includes the following figure supplement(s) for figure 5:

**Figure supplement 1.** Comparison of RNA Pol two distribution over the *opa* and *ftz* loci.

**Figure supplement 2.** Effect of *opa* loss- and gain-of-function across all ATAC peaks.

**Figure supplement 3.** Enrichment of Zld, Bcd, and Opa dependence in accessibility classes.

**Figure supplement 4.** Expression of *tub >opa* compared with wild-type *opa*.

targets (*Figure 5C*). We first determined a set of high-confidence direct Opa-binding sites by performing ChIP-seq on an engineered allele of *opa* in which we introduced by CRISPR a *3x-myc* epitope tag into the 3' end of the *opa* coding region. The resultant *opa-myc* allele is homozygous viable, is expressed within the expected domain (*Figure 5A*) with expected kinetics, and has no detectable adverse effects from engineering of the *opa* locus (data not shown). We performed ChIP-seq on three independent biological replicates of 200 homozygous *opa-myc* cellular blastoderm embryos using as a negative control wild-type ($w^{1118}$) embryos. Mapped reads were subjected to peak calling with MACS2 (*Zhang et al., 2008*), yielding 3553 total, low-stringency peak regions. From these, we we defined a set of 879 reproducible, high-confidence Opa ChIP peaks by Irreproducible Discovery Rate analysis (*Landt et al., 2012*) (see Materials and methods and *Supplementary file 3* for the IDR filtered peaks list). 85% (744/879) of the Opa ChIP peaks overlap with at least one ATAC-seq peak. Opa ChIP peaks contain between 0 and 9 Opa motifs (80% match to the Opa position weight matrix reported by MEME), with an average of 1.08 motifs per peak. To test whether Opa pioneers accessibility at its direct targets, we performed ATAC-seq on single NC14 + 72' embryos collected from a cross between parents heterozygous for a null allele (*w; opa^{IIP32}/His2Av-GFP*). Following mapping of reads, zygotic genotypes were called based on recovered SNPs (see Materials and methods). DESeq2 analysis of differentially accessible regions between genotypes determines that 263 (30%) of the high-confidence Opa-binding sites show significantly reduced chromatin accessibility in homozygous *opa*-mutant embryos (*Figure 5C*).

We have chosen in the following to focus on changes in chromatin accessibility specifically over the set of direct Opa-binding sites as determined by ChIP-seq to help distinguish between direct and indirect effects of Opa on the system. Evaluating changes in accessibility instead over the entire ATAC-seq peaks list, we find that 319 peaks lose, and 26 peaks gain accessibility in *opa* mutant samples (*Figure 5—figure supplement 2*). None of the 26 peaks that gain accessibility in *opa* mutant samples overlap with either high- or low-stringency Opa ChIP peaks and these sequences are enriched neither for an Opa nor any other motif (data not shown). This suggests that these *opa*-dependent gains in accessibility are indirect effects. Of the set of 319 peaks with

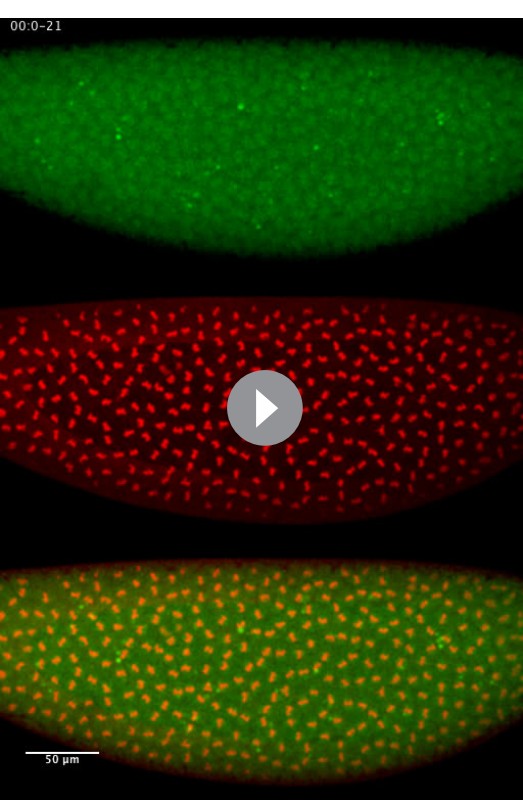

**Video 1.** Representative Movie of Opa-llama expression. An embryo expressing maternal free EGFP, maternal Histone H2Av-RFP, and zygotic llama-tagged *opa* is shown here in a time-lapse image that initiates at mitosis 12 and continues through early germband extension. The counter in the upper left corner indicates time (hh:mm) relative to the start of NC14. Top panel shows the opa-llama (EGFP) channel. Middle panel shows His2Av-RFP, which marks nuclei at all times. Bottom panel shows the merge of these two channels. In addition to features of the movie described in the Results section, note the persistence of Opa on mitotic chromatin during zygotic mitoses within the segmental primordium (>1 hr:15 m) compared with EGFP background in head region mitotic domains.

https://elifesciences.org/articles/53916#video1

reduced accessibility, 139 (44%) overlap with high-stringency, IDR-selected Opa ChIP peaks. Testing for overlap with the low-stringency, pre-IDR Opa ChIP peaks list, an additional 63 (20%) Opa peaks overlap. The remaining 167 peaks that demonstrate reduced accessibility in an *opa* mutant are not enriched for an Opa motif, suggesting that at least 37% of the overall effect of Opa on accessibility gains is indirect. Motif enrichment on this set of 167 indirect Opa-sensitive peaks yields a long, adenine-rich motif with no high-confidence match in the sequence motif database and a weak resemblance to a binding motif for *jim,* another C2H2 Zn-finger transcription factor (data not shown). Since *jim* is not expressed until significantly later in fly development (>12 hr post-fertilization, primarily in larval and pupal stages), we conclude that the factor associated with this motif is unknown.

These results demonstrate that Opa is necessary for conferring accessible chromatin states at a subset of its direct binding sites. Similar to the overall Opa peaks list, opa-dependent ChIP peaks contain between 0 and 9 Opa motifs, but have moderately higher average motif number (1.32). In contrast, ChIP peaks that do not depend on Opa for accessibility have fewer Opa motifs (range: 0 to 5, mean = 0.95). This moderate difference in average motif content between these two classes is statistically significant at $p=2\times10^{-7}$ by one-tailed permutation test on $1 \times 10^7$ proportional randomly selected groups. Similar to the overall distribution of dynamic chromatin regions from ZGA to gastrulation, opa-dependent regions are strongly enriched for intergenic regions and are under-represented for promoter regions. Only three Vienna Tiles, all with late NC14 AP-stripe expression patterns, uniquely overlap with the set of opa-dependent sites: VT14361 (contains *eve-late*), VT1965 (contains *slp1 '5'*), and VT39542 (intronic *homothorax* enhancer). Relative to the set of patterning-dependent and independent dynamic genomic regions defined above, direct opa-dependent pioneer activity alone is sufficient to account for 36% of group 9 (*Figure 4*): the set of chromatin regions that depend on inputs from maternal patterning systems to gain accessibility by the onset of gastrulation (*Figure 5—figure supplement 3*). Therefore, in addition to pioneering open chromatin in late NC14 embryos, Opa's zygotic pioneer activity accounts for a significant proportion of overall changes in chromatin accessibility downstream of maternal patterning systems.

In addition, Opa is sufficient to pioneer open chromatin at the majority of these direct, opa-dependent targets. Because *opa* is expressed late in NC14, at earlier timepoints opa-dependent regions represent effectively naive chromatin states that can be used to test for sufficiency. We reasoned that at NC13 (one cell cycle before NC14) chromatin accessibility status at Opa targets would largely resemble late *opa*-mutant states. To test for sufficiency, we performed ATAC-seq comparing single wild-type embryos collected 12 min into NC13 with embryos misexpressing *opa* maternally under direct control of the maternal *alpha-tubulin 67* c promoter (*tub >opa*) (*Figure 5C*). This misexpression strategy yields maternal *opa* expression levels comparable to maximal zygotic *opa* expression at late NC14 (*Figure 5—figure supplement 4*). Embryos produced from mothers heterozygous for *tub >opa* hatch and can mature to adulthood, albeit with varying degrees of segmental mis-patterning as described below. Whereas wild-type NC13 + 12' accessibility at direct Opa targets is nearly indistinguishable from NC14 + 72' *opa*-mutant samples, maternal expression of Opa increases the open chromatin signature at these targets at 80% (197/263) of opa-dependent sites and at 59% (519/879) of direct Opa targets overall. We conclude that, with the exception of 25% (66/263) opa-dependent but maternal-opa-insensitive sites that Opa is sufficient to pioneer open chromatin at its target sites.

A second criterion for classifying a transcription factor as a pioneer is that it binds to its DNA motifs even in the nucleosome-associated state (*Iwafuchi-Doi and Zaret, 2016*). To test this, we measured the distribution of Opa-binding motifs relative to nucleosome positions modeled from the distribution of large (>100 bp) ATAC-seq fragments (*Schep et al., 2015*). Either before expression of *opa* (wild-type NC14+12') or in *opa* mutants at late NC14 (*opa* NC14 + 72'), 51% (250/486) of Opa motifs in opa-dependent regions are located within 73 bp of a modeled nucleosome dyad (i.e., within the wrap of DNA around a nucleosome, *Figure 5D* and data not shown). The fraction of Opa motifs within opa-dependent regions overlapping with predicted nucleosome positions is significantly greater than is observed from the set of Opa motifs located in non-bound, but accessible regions (27%, 427/1572, *Figure 5D*). Following binding of Opa (wild-type NC14+72'), the fraction of motifs in bound regions overlapping the nucleosome footprint reduces to 15% (73/486) and the average position of Opa motifs relative to predicted nucleosome dyads increases by an average of 79 bp in bound regions (*Figure 5D*). In contrast, the distance between motifs and nucleosomes distance changes by only 6 bp in non-bound regions. These results suggest that a substantial fraction

of Opa motifs within future Opa-occupied sites are associated with nucleosomes in the ZGA chromatin state. Following activation of Opa late in NC14, binding of Opa to targets results in a reorganization of nucleosomes surrounding Opa motifs and exposure of previously nucleosome-occluded DNA. Although additional studies of Opa binding to nucleosome-free and -bound DNA will need to be performed, these results are consistent with a model where Opa can interact with nucleosome-associated binding motifs and can trigger reorganization of local chromatin structure.

## Additional mechanisms regulate developmental competence of late segment-polarity CRMs to gain accessibility

We have demonstrated that following ZGA, both local and global changes in chromatin accessibility patterns continue to take place. The identification of pioneers like Opa raises the possibility that distinct zygotic factors function to establish accessibility states conditional on prior, maternal patterning information. We wished to investigate the hypothesis that the sequence of chromatin accessibility changes are themselves critical for the proper execution of the developmental patterning program. To address this, we therefore quantified opa-dependent chromatin accessibility within the segmentation network and evaluated the consequences of premature *opa* expression on patterning. We predicted that maternal mis-expression of *opa* would effectively conflate a ZGA chromatin state with a gastrulation chromatin state. Opa is necessary for full chromatin accessibility at a set of late pair-rule and segment polarity CRMs (*Figure 6A–C*). Late-acting opa-dependent CRMs within the pair-rule network include the late *eve* seven-stripe element (*eve-late* also referred to as *eve-autoregulatory*,(*Harding et al., 1989*; *Goto et al., 1989*; *Fujioka et al., 1996*), see also *Figure 6—figure supplement 1*), the 'center cell' repressor that splits early *prd* stripes into anterior and posterior stripes (*prd cc repressor Gutjahr et al., 1994*), a late anterior stripe repressor for *prd* (*prd A8 repressor Gutjahr et al., 1994*), several late *slp1* enhancers (*u3525, i1523, u4739,* and '5', (*Fujioka and Jaynes, 2012*; *Sen et al., 2010*) see also *Figure 6—figure supplement 1*), as well as a novel *odd* late enhancer (*odd-late, Figure 6B,D and E*). Several segment polarity CRMs are also opa-dependent, including a *gsb* 3' enhancer (*Gurdziel et al., 2015*, *Figure 6C*), a portion of the *en H* regulatory element (*en H2* (*Cheng et al., 2014*), see *Figure 1*), an *en* intronic enhancer (*Cheng et al., 2014*) as well as less well characterized elements within *ptc* and *wg* defined by large-scale enhancer screens (*ptc 5'*=GMR69 F07, *ptc 5'* (2)=GMR69 F06, *wg intron* = GMR16 H05)(*Jenett et al., 2012*). We confirmed an effect of *opa* loss of function on the expression patterns of three opa-dependent CRMs, *odd-late, eve-late,* and *slp1 '5'* (*Figure 6B,D* and *Figure 6—figure supplement 1*). In *opa* mutants, no expression is seen from *odd-late* (*Figure 6D*). The effect of *opa* on *eve-late* is also nearly complete, although reduced levels of expression persist within stripe 1 (*Figure 6—figure supplement 1*). Loss of *opa* also reduces expression from *slp1 '5'*, completely eliminating activity within secondary odd parasegmental stripes, and significantly reducing activity within the primary even parasegmental stripes (*Figure 6—figure supplement 1*). These effects are largely consistent with the range of previously reported *opa* loss of function phenotype on pair-rule and segment polarity targets (*Clark and Akam, 2016*) and supports the conclusion that the primary function of *opa* in the segmentation network is to modulate the temporally restricted accessibility of a subset of critical CRMs.

Notably, although *opa* is sufficient to induce accessible chromatin at most of the late pair-rule opa-dependent targets (*odd-late, eve-late, slp1 '5', slp1 i1523, prd 01, prd A8 repressor*) as well as three regions within the *opa* locus itself, segment polarity targets show a distinctly reduced sensitivity to gain accessibility in response to premature *opa* expression, with only *en H2* and an intronic *wg* region having marginally above-threshold increased accessibility. To characterize this further, we performed motif enrichment analysis over the entire set of opa-dependent, opa-insensitive sites and find enrichment of the Opa motif, as well as enrichment of two maternal repressors, Ttk and Cg. In contrast, the set of both opa-dependent and opa-sensitive regions shows enrichment for the Opa motif alone. These results indicate that within the segmentation network there exists differential sensitivity to acquisition of chromatin accessibility states, and that although Opa is necessary for conferring open chromatin at a set of pair-rule and segment polarity loci, it is possible that competence to respond to Opa, and perhaps any pioneer in general, is further regulated by additional repressive mechanisms.

Maternal expression of *opa* triggers premature activity of opa-dependent targets and mispatterning of the pair-rule network (*Figure 6D–F*). The opa-sensitive target *odd-late* initiates expression in wild-type embryos in a domain of seven odd-numbered parasegments that correspond to the

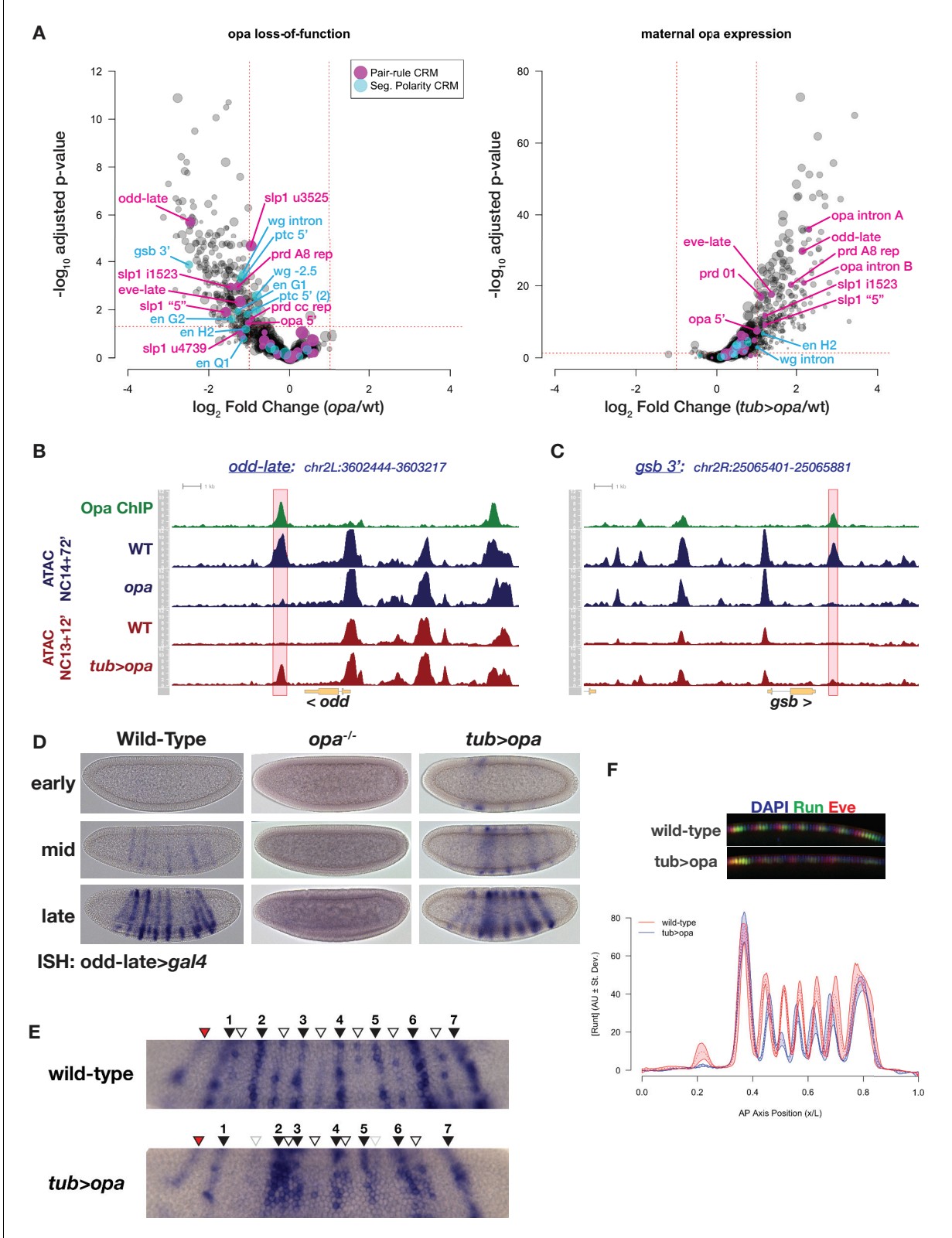

**Figure 6.** Premature expression of Opa disrupts pair-rule patterning. (**A**) Volcano plots for loss of function (left) and maternal misexpression (right) of Opa with pair-rule and segment polarity CRMs highlighted (magenta and cyan, respectively). (**B**) Example of ATAC seq coverage over an opa-dependent pair-rule locus, *odd*. The *odd-late* CRM is highlighted showing the effect of *opa* at this element. The scale for the *y*-axis is 0–12 CPM for all plots. (**C**) For comparison, ATAC-seq coverage over a segment polarity locus, *gsb*, is shown. The *gsb* 3′ CRM is highlighted. The scale for the *y*-axis is

*Figure 6 continued on next page*

*Figure 6 continued*

0–15 CPM for all plots. (D) In situ hybridization for an *odd-late gal4* reporter is shown for wild-type, *opa* mutant and *tub >opa*. Embryo stages are indicated at left. Note the lack of activity in *opa* mutants and the premature activation in the presence of tub >opa. (E) Detail view of *odd-late* expression in wild-type and *tub >opa* gastrula stage embryos. Odd parasegmental stripes are indicated with numbered black arrowheads, even stripes with open arrowheads, and the anterior head stripe is indicated with a red arrowhead. Weak even parasegmental stripes in *tub >opa* that eventually appear are indicated with grey open arrowheads. The stripe at numbered position 1 is coincident with the cephalic furrow, which is beginning to form in both pictured embryos. (F) Maternal *opa* interferes with stripe positioning and intensity for pair-rule genes *eve* and *runt*. Plot shows the average effect of maternal *opa* on *runt* expression in mid NC14 embryos. Average quantified Runt expression ±std. dev. is plotted for wild type (red) and *tub >opa* (blue, n = three embryos per genotype). Inset shows dorsal mid-saggital view of a representative embryo of the indicated genotypes stained for Runt (green) and Eve (red). See also *Figure 6—figure supplement 1*.

The online version of this article includes the following figure supplement(s) for figure 6:

**Figure supplement 1.** Examples of expression of opa-dependent CRMs in wild-type and opa mutant embryos.

secondary pattern of *odd* expression (*Figure 6D,E*; *Clark and Akam, 2016*). Expression within even-numbered parasegments corresponding to the primary *odd* expression pattern follows later, ultimately yielding a 14-stripe pattern (*Figure 6E* and data not shown). In addition to the parasegmental expression pattern, there is a transient head stripe anterior to the first stripe (*Figure 6E*). Activity of this CRM is entirely dependent on *opa* (*Figure 6D*). Maternal expression of *opa* affects both spatial and temporal aspects of the *odd-late* expression pattern, driving premature activity of *odd-late* in stripes with incorrect spatial distributions. The most significant spatial effect of maternal *opa* expression on *odd-late* pattern is an increased interstripe distance between the first and second odd para-segmental expression domain, resulting in apparent compression between the second and third stripes (*Figure 6E*). In addition, compared with wild-type, the regular spacing of odd- and even-par-asegmental stripes at positions 4–6 is disrupted with maternal *opa* expression. Despite the fact that *tub >opa* is expressed uniformly across the entire embryo, premature *odd-late* expression appears not uniformly, but within the spatial domains to which its later expression will be restricted (*Figure 6D*). Similar effects have been observed with premature expression of *opa* using the *gal4*-UAS system, where increasing levels of *opa* expression lead to defects in *slp1* expression only within the segmental primordium (i.e. in a domain where additional co-regulators are expressed). Only in combination with uniformly misexpressed *runt* are ectopic domains of *slp1* observed outside the segmental primordium (*Swantek and Gergen, 2004*). These observations are consistent with a role for *opa* largely restricted to regulating accessibility and not strictly activating or repressing expression from direct targets such as *odd-late* and that interfering with timing of CRM accessibility results in inappropriate responses to developmental cues and disruption of the precision of embryonic patterning.

To quantify the effect of maternal *opa* expression on additional components of the pair-rule network, we immunostained embryos for Runt and Eve and quantified stripe positioning in precisely staged embryos (*Dubuis et al., 2013*). Although between genotypes the segmental primordium is unchanged in both overall area and overall positioning, the intensity and positions of stripes 2–6 are affected by maternal *opa* expression. Similar to *odd-late* we also observe increased inter-stripe distances between Runt stripes 1 and 2, and reduced distance between stripes 2 and 3, as well as minor mis-positioning of stripes 4–6. We note that the long-term severity of the *tub >opa* phenotype may be attenuated by limited effects of maternal *opa* expression on the segment polarity network. Because the degree of *opa* mis-expression approximates wild-type *opa* expression levels at mid to late NC14 (*Figure 5—figure supplement 4*), it also remains possible that driving higher levels of *opa* expression would result in stronger mis-expression phenotypes. The positioning of stripes in the pair-rule network is highly precise and results from optimal decoding of maternal and gap-gene patterning inputs (*Petkova et al., 2019*). Our results are consistent with a model where premature *opa* expression alters how maternal information is read at the level of this network. While we cannot at present distinguish effects on the system that stem from Opa-dependent pioneering from possible Opa-dependent effects on transcription, these results provide support for the hypothesis that the sequential transitions in chromatin accessibility patterns are themselves necessary for proper patterning of the embryo.

## Discussion

To the extent that the developmental program can be described by gene regulatory networks, mechanisms for the regulation of chromatin accessibility must play a major role in determining how this program unfolds. Focusing on the AP segmentation network, we show that at the outset of embryonic patterning, not all CRMs in this network are fully accessible. Over time, accessibility states change as the network activates the gap, pair-rule, and segment polarity components of the system. A portion of these changes over time are sensitive to loss of maternal patterning cues, which either directly (via maternal factors such as Bcd *Hannon et al., 2017*) or indirectly (via activation of zygotic targets such as Opa) influence accessibility states. Therefore, accessibility states within a network can be dynamic, and the drivers of these changes can be integral components of the networks themselves, as we see in the case of the segmentation GRN. As gene expression states progress through a stereotypical progression in order to impart embryonic pattern, so do accessibility states of the CRMs that drive these patterns. Importantly, we demonstrate that the temporal sequence of CRM accessibility states is itself important for proper patterning of the embryo: driving premature accessibility at a subset of late pair-rule CRMs results in inappropriate responses to patterning cues, and disruption to the otherwise highly precise expression of pair-rule genes.

Although we have focused primarily on the AP segmentation network and the contribution of Opa to accessibility of late-acting enhancers in this network, we have also more generally estimated the relative contribution of patterning-dependent and patterning-independent mechanisms for driving changes in chromatin accessibility in the 1-hr period between ZGA and gastrulation. The ATAC-seq peaks in this study have been provided as an extended data file with complete annotations for further exploration of these results. We caution that we have not exhaustively determined here the extent of patterning-dependent accessibility states on the DV network (see below). In addition to CRMs within the segmentation network, our results also highlight at least two other major classes of dynamic chromatin sites.

First, regions that demonstrate enriched chromatin accessibility in mutant *bcd osk cic tsl Tl^{RM9}* embryos compared with wild-type are enriched for CRMs that demonstrate expression patterns restricted to cells fated to become posterior endoderm. This is a lineage that has received comparatively less attention in studies of early *Drosophila* development in part because defects in endodermal specification were not explicitly screened for in the classic zygotic screens for patterning mutants (*Wieschaus and Nüsslein-Volhard, 2016*). Similarly, a recent ATAC-seq study on spatially restricted populations of blastoderm nuclei did not exclusively purify cells within this lineage (*Bozek et al., 2019*). Our results indicate that this small population develops extensive cell-type specific accessibility patterns by the onset of gastrulation, and we speculate that our data may enrich the set of known CRMs operating within this lineage. By motif enrichment analysis within regions that gain accessibility late in mutant samples (*Figure 4*, group 10), we find over-representation of both Fkh and Byn motifs. In addition to the well-known role for mammalian Fkh homologs in pioneering endodermal chromatin states (*Iwafuchi-Doi et al., 2016*; *Cirillo et al., 2002*; *Ang et al., 1993*), a recent study has also indicated that the mammalian Byn homolog, Brachyury, contributes to pioneering of mesendodermal enhancers during differentiation of embryonic stem cells (*Tosic et al., 2019*). Therefore, in addition to potentially identifying additional components of a *Drosophila* posterior endodermal GRN, our results also suggest that mechanisms for establishing endodermal cell-type specific chromatin states are conserved. Further work will be needed to confirm the expected pioneer activities of Fkh and Byn in the *Drosophila* system.

Second, our observation of continued uniform, patterning-independent changes to chromatin accessibility after ZGA suggests that additional global timers of developmental progression continue to operate following the maternal-to-zygotic transition. One of the most intensely studied developmental transitions in chromatin structure is large-scale ZGA that accompanies the maternal-to-zygotic transition. The initial ground state of chromatin structure is built by maternally supplied pioneers, particularly Zelda (*Blythe and Wieschaus, 2016*; *Liang et al., 2008*; *Nien et al., 2011*; *Sun et al., 2015*; *Harrison et al., 2011*; *Schulz et al., 2015*), and likely a combination of factors that bind $(GA)_n$ repeats, GAGA-factor and CLAMP (*Blythe and Wieschaus, 2016*; *Rieder et al., 2017*; *Blythe and Wieschaus, 2015*; *Kaye et al., 2018*). Prior studies on the temporal dynamics of the establishment of the initial ground state have suggested that these pioneers act in distinct temporal waves, with the earliest accessible regions associated with Zelda binding, and the latest accessible

regions enriched for the $(GA)_n$ motif and GAGA-factor binding (*Blythe and Wieschaus, 2016*). Our results add to these prior observations by suggesting that these global 'waves' of accessibility regulation continue well past the ZGA, and likely, based on motif enrichment, receive inputs from additional maternally supplied factors with expected repressive activity: Ttk, Aef1, and Cg. Next to nothing is known about the role of these factors specifically in the context of global chromatin accessibility regulation at the ZGA. Aef1 was identified as a Zn-finger transcriptional repressor that regulates gene expression in adult *Drosophila* fat body (*Falb and Maniatis, 1992*). Cg has been implicated in both positive and negative regulation of target genes, has been shown to interact with GAGA factor (*Lomaev et al., 2017*), and plays a role in recruitment of polycomb group proteins to polycomb response elements through direct binding of $(CA)_n$ repeats (*Ray et al., 2016*). Both Aef1 and Cg are expressed maternally, and transcripts are cleared by the onset of gastrulation (Berkeley *Drosophila* Genome Project) although protein expression kinetics have not been determined for early embryos. The transcriptional repressor Ttk plays a role in the spatio-temporal regulation of segmentation gene expression, directly influencing timing and pattern of pair-rule and segment polarity gene expression (*Pritchard and Schubiger, 1996*; *Harrison and Travers, 1990*; *Brown and Wu, 1993*; *Read et al., 1992*; *Wheeler et al., 2002*). Perhaps indicating a more general role beyond segmentation network regulation, Ttk has been demonstrated to interact directly with GAGA factor and antagonize GAGA-dependent transcriptional activation (*Lomaev et al., 2017*; *Pagans et al., 2004*). Because Ttk mRNA and protein are expressed at high levels maternally and are cleared from the embryo by gastrulation (*Harrison and Travers, 1990*), Ttk (as well as Aef1 and Cg) may play a more global role in ZGA timing by limiting pioneer factor activity at target sites until they are cleared from the embryo. In this respect, it is interesting to note that the subset of opa-dependent targets that are insensitive to maternal *opa* expression demonstrate an enrichment for Ttk and Cg motifs. One possible explanation for this apparent developmental competency to respond to Opa pioneering activity is that binding of maternal repressors can antagonize pioneer factor activity. Future work will include testing the role of these maternal factors in the context of ZGA timing and regulation of coordinated, global chromatin remodeling events.

## Odd-paired as a pioneer within the segmentation network

In contrast with these mechanisms for uniform regulation of accessibility, while there is relatively little influence of maternal patterning systems directly on chromatin accessibility status, we observe that certain zygotic targets of maternal pathways, such as Opa, can have a major impact on chromatin accessibility states. Similar to Zelda (*Schulz et al., 2015*), Opa is necessary for driving accessibility at ~30% of its direct binding targets. We note that overall Opa displays more limited binding across the *Drosophila* genome and occupies a smaller number of available motifs encoded in the genome sequence than Zelda (*Harrison et al., 2011*). The reason for this is not clear. Additionally, we demonstrate evidence consistent with a model where Opa can bind to motifs in the nucleosome bound state (*Figure 5D*), although this result will need to be confirmed through future biochemical studies of Opa binding. On the basis of these two observations, that Opa is necessary and sufficient for driving open chromatin states, and that Opa likely interacts with inaccessible motifs to drive these states, we conclude that Opa functions as a pioneer factor in this system.

Opa's primary role in the pair rule network is to facilitate the transition, termed a 'frequency doubling', from early to late expression patterns. In the absence of Opa, pair-rule loci (primarily *odd*, *slp*, *run*, and *prd*) fail to undergo the transition from early seven-stripe to late 14-stripe patterns. Additionally, late 7-stripe expression of *eve* is also strongly affected (*Clark and Akam, 2016*). Because of uniform expression across the segmental primordium, Opa does not provide positional information that defines the precise location of its target expression domains (*Benedyk et al., 1994*), and has been proposed to cooperate with additional pair-rule factors such as Runt to activate or repress target gene expression (*Clark and Akam, 2016*; *Swantek and Gergen, 2004*). Here, we demonstrate the mechanism for Opa's role in the network: that Opa facilitates the frequency doubling of the pair-rule network by pioneering accessibility of the CRMs that drive these late expression patterns. We predict that Opa pioneer activity will therefore result in conditional cis-regulatory interactions of the remaining pair-rule factors with late CRMs. This mechanism can help explain the previously observed 'conditional regulation' between network components (e.g. Odd repression of *prd* to yield anterior and posterior stripes)(*Clark and Akam, 2016*), which we propose is largely mediated through opa-dependent CRM accessibility states. The set of opa-dependent CRMs within

the pair-rule network that we identify strongly support this conclusion. Incorporating such 'time-gated' pioneering events into a regulatory network may therefore allow for a system to generate multiple patterning outputs from a limited set of input transcription factors. Further investigation of the opa-dependence for conditional cis-regulatory interactions amongst pair-rule factors, as well as identification of additional zygotic pioneer factors will address these predictions.

A critical distinction that arises between transcription factors within a network, then, is what effect they have on chromatin accessibility states. It is likely that not all transcription factors have pioneer activity, or that the ability of a factor to pioneer is context specific. For instance, loss of *grainyhead* (*grh*) has minimal effects on the pre-gastrula chromatin accessibility state, despite the fact that *grh* has been demonstrated to function as a pioneer in other biological contexts (*Nevil et al., 2019*). Similarly, while repressors have been demonstrated to negatively impact chromatin accessibility states, certain repressors, such as the pair-rule factor *hairy*, can operate not through compaction of chromatin but by inhibiting recruitment of the basal transcriptional machinery, at least in certain contexts (*Li and Arnosti, 2011*). Whether repressors within the pair-rule network fall into distinct chromatin-dependent and -independent categories at a genome-wide scale remains to be determined. A comprehensive appraisal of how transcription factors within a network not only interact with cis-regulatory elements over time but also how they impact chromatin accessibility states will be necessary to fully understand the regulatory logic of embryonic patterning.

## Is accessibility regulated maternally along the DV axis?

We note that we have not yet exhaustively examined all possible maternal patterning contributions to chromatin accessibility. The *bcd osk cic tsl Tl^{RM9}* mutant embryos used in this study, while amorphic for maternal determinants of AP and terminal cell fates, have uniform, moderate *Tl* pathway activation and therefore moderate *dorsal* (*dl*) activity (*Roth et al., 1989*). So, it remains possible that any possible *dl*-dependent chromatin accessibility states remain unidentified by our study. However, we predict that *dl* does not pioneer chromatin at least to the same extent as *bcd*. The observation of *bcd*-dependent chromatin accessibility states has now been observed in four independent studies in addition to this work (*Cusanovich et al., 2018a*; *Hannon et al., 2017*; *Haines and Eisen, 2018*; *Bozek et al., 2019*). Besides studies where accessibility was measured in *bcd* mutants directly (this work and *Hannon et al., 2017*), two additional studies of spatially restricted chromatin accessibility independently confirmed the predicted (*Hannon et al., 2017*) anterior enrichment of *bcd*-dependent regions (*Haines and Eisen, 2018*; *Bozek et al., 2019*). However, none of these studies were specifically designed to distinguish differential DV accessibility states. In contrast, a recent single-cell ATAC-seq study could have identified these states if they existed (*Cusanovich et al., 2018a*). This study consistently identified several clusters of early embryonic cells that distinguished AP states, and single-cell 'anterior' clusters were found to be enriched for previously identified *bcd*-dependent regions (*Cusanovich et al., 2018a*). If *dl* were to pioneer chromatin to a similar extent as *bcd*, we would therefore expect that single-cell ATAC-seq would have also identified early 'dorsal' and 'ventral' clusters, but in this study, all DV-specific clusters (e.g. mesoderm) were only found associated with cells presumed to be staged later based on 'pseudotime' analysis. So, although maternal systems may not drive differential accessibility along the DV axis, it nevertheless likely that, similar to *opa*, specific zygotic targets within the DV networks operate immediately downstream of maternal pathways to distinguish DV-specific chromatin accessibility states. This also highlights how in the *Drosophila* system both single-cell (or enriched cell) methods and genetic manipulation represent powerful complementary approaches for distinguishing the degrees of complexity in the chromatin landscape and linking these features to developmental regulatory systems. Replacing *Tl^{RM9}* with complete gain- or loss- of function alleles affecting *Tl* signaling (*Stathopoulos et al., 2002*) in our maternal mutants will definitively test the hypothesis suggested by single-cell approaches of limited early DV heterogeneity in chromatin accessibility states.

## Transcriptional activation or pioneering?

A prior study (*Bozek et al., 2019*) observed that chromatin accessibility levels at enhancers as measured by ATAC-seq correlated strongly with the transcriptional activity of the associated gene. This raises important questions about how to interpret ATAC-seq peak intensities, including whether losses of ATAC signal in certain mutant backgrounds can be interpreted as stemming from pioneer

activities or simply reflect reduced expression of associated genes. While this observation points to likely several compounding layers of regulation that influence chromatin accessibility and ATAC-seq measurements, it is unlikely that transcriptional activity is the sole determinant of enhancer accessibility states. For one, sites can be found that gain accessibility uniformly, but which are not transcribed. One such locus, *ush*, undergoes significant patterning-independent gains in promoter accessibility between ZGA and gastrulation (*Figure 4—figure supplement 2*), but fails to be expressed in the mutant (*Figure 2*). This demonstrates that gains in accessibility can occur independently of transcriptional activation, and similarly highlights an example where transcriptional repression need not result in reduced accessibility. In contrast, the high degree of ATAC signal we observe at sites that preferentially gain accessibility in uniform-lineage mutant embryos (e.g. *wg* −1 (*Figure 3A,B*) and *18w 1* (*Figure 4—figure supplement 10*)) may be significantly compounded by the large fold-increase between wild-type and mutant embryos of cells in which the associated CRM drives active transcription (*Figure 3B*). So, we cannot rule out completely an additional contribution from transcriptional status on the magnitude of ATAC signals.

In our study, we demonstrate distinct patterning-dependent gains in chromatin accessibility over time, and a significant fraction of these we link to *opa* activity. However, within the segmentation network specific CRMs, Opa is not the sole pair-rule factor predicted to bind to these sites. At the *odd-late* enhancer, for instance, in addition to our demonstration of direct Opa binding, prior ChIP-chip profiling (*MacArthur et al., 2009*) of a subset of pair rule genes in broadly staged embryos predicts that one activator (Ftz) as well as several repressors (Slp1, Prd, Run, and to a lesser extent Hairy) bind to this region. Ftz, Run, and Hairy are expressed during early NC14, well before Opa, yet neither is *odd-late* functional at this time, nor is it accessible. Our results are consistent with a model where this and other *opa*-dependent enhancers require Opa to pioneer accessible chromatin first in order for other regulatory factors such as Ftz and Run to interact with the underlying DNA sequence and thereby regulate patterned expression of the associated gene. It is possible that robust transcriptional activity following initial pioneering results in additional increases in overall ATAC signal. Future work will include directly testing whether Opa expression facilitates de novo interaction of pair-rule genes at previously inaccessible chromatin regions, and direct testing of the regulatory consequences of dynamic chromatin accessibility states on network function.

We concede that it is difficult to disentangle using ATAC-seq alone what are likely numerous additional determinants of developmental chromatin accessibility dynamics. For instance, ATAC does not provide good information about the chromatin state of enhancers prior to acquiring an accessible state, nor does it directly shed light on the mechanisms for regulating competence for a site to be pioneered. In the case of Opa, we identify shifts in the positions of nucleosomes relative to binding sites consistent with a model where Opa binding triggers reorganization of chromatin structure at targeted loci. In many, but not all cases, Opa is sufficient to open chromatin at these sites earlier in development. Additional contextual features of the system must regulate competency for a site to be pioneered. As observed previously (*Haines and Eisen, 2018*; *Bozek et al., 2019*), in many cases regions that undergo spatially restricted, pattern-dependent changes in chromatin accessibility demonstrate intermediate degrees of accessibility in the 'closed' state greater than that observed at completely inactive genomic loci. The idea of distinct degrees of 'repressed' or 'closed' chromatin adds further layers of potential regulation to these systems and raises the possibility that additional factors confer competence for chromatin to be pioneered. Much remains to be understood about what grants competence to a genomic locus to undergo chromatin remodeling in response to binding of a pioneer factor.

Finally, we note that a contemporaneous and independent investigation on the effect of Opa on early embryonic chromatin states has also been reported (*Koromila et al., 2019*).

# Materials and methods

**Key resources table**

| Reagent type (species) or resource | Designation | Source or reference | Identifiers | Additional information |
|---|---|---|---|---|
| Genetic reagent (*D. melanogaster*) | *bcd^{E2} osk^{166} cic^1 tsl^1 Tl^{RM9}* | This paper | Flybase allele IDs: FBal0001081 FBal0013308 FBal0103875 FBal0017195 FBal0016839 | Quintuple mutant generated by multiple rounds of meiotic recombination as described. |
| Genetic reagent (*D. melanogaster*) | *opa^{IIP32}* | *Jürgens et al., 1984* | Flybase allele ID: FBal0013272 | |
| Genetic reagent (*D. melanogaster*) | *P(His2Av-EGFP)* | Bloomington *Drosophila* Stock Center | Flybase allele ID: FBal0183310 | |
| Genetic reagent (*D. melanogaster*) | *opa-3xMyc* | This paper | | CRISPR/Cas9 allele |
| Genetic reagent (*D. melanogaster*) | *opa-EGFP llama* | This paper | | CRISPR/Cas9 allele |
| Genetic reagent (*D. melanogaster*) | Bcd promoter > EGFP | This paper | | Transgenic insertion into P(CaryP)attP40 (Flybase transposable element insertion site ID: FBti0114379) |
| Genetic reagent (*D. melanogaster*) | *alpha Tubulin 67c > opa-3xMyc* | This paper | | Transgenic insertion into P(CaryP)attP40 (Flybase transposable element insertion site ID: FBti0114379) |
| Antibody | Anti-myc polyclonal antibody (rabbit) | Sigma-Aldrich | Cat #: C3956 | ChIP: 1 ug IF: 1:200 |
| Antibody | Anti-eve monoclonal antibody (mouse) | Developmental Studies Hybridoma Bank | Clone number 2B8 | IF: 1:20 |
| Antibody | Anti-runt polyclonal antibody (guinea pig) | Wieschaus Lab, Princeton University | | IF: 1:1000 |
| Software, algorithm | TrimGalore! | https://github.com/FelixKrueger/TrimGalore | | |
| Software, algorithm | Bowtie2 | *Langmead and Salzberg, 2012* | | |
| Software, algorithm | Picard MarkDuplicates | http://broadinstitute.github.io/picard/ | | |
| Software, algorithm | MACS2 | *Zhang et al., 2008* | | |
| Software, algorithm | IDR | https://www.encodeproject.org/software/idr/ | | |
| Software, algorithm | Bedtools | https://bedtools.readthedocs.io/en/latest/ | | |
| Software, algorithm | Samtools | http://www.htslib.org/ | | |
| Software, algorithm | GenomicAlignments 1.18.1 | *Lawrence et al., 2013* | | |

*Continued on next page*

*Continued*

| Reagent type (species) or resource | Designation | Source or reference | Identifiers | Additional information |
|---|---|---|---|---|
| Software, algorithm | GenomicRanges 1.34.0 | *Lawrence et al., 2013* | | |
| Software, algorithm | DESeq2 1.22.2 | *Love et al., 2014* | | |
| Software, algorithm | Gviz 1.26.5 | *Hahne and Ivanek, 2016* | | |
| Software, algorithm | Biostrings 2.50.2 | *Pagès et al., 2019* | | |

## *Drosophila* stocks and husbandry

All fly stocks were maintained on an enriched high-agar cornmeal media formulated by Gordon Gray at Princeton University. Embryos were collected on standard yeasted apple-juice agar plates mounted on small cages containing no more than 300 adults aged no older than 14 days. Media compositions are available upon request.

The wild-type stock for ATAC-seq experiments is *w; His2Av-EGFP* (III), and is *w$^{1118}$* for immunostaining and in situ hybridizations. The quintuple 'patternless' maternal mutant was a gift from Eric Wieschaus at Princeton University. The base stock genotype is *bcd$^{E2}$ osk$^{166}$ cic$^1$ tsl$^1$ Tl$^{RM9}$* and was constructed through multiple rounds of meiotic recombination. A commonly available *bcd osk* double mutant stock was recombined with *cic tsl* double mutants and resultant quadruple mutants were subsequently recombined with *Tl$^{RM9}$*. Retention of mutant alleles was confirmed at each step by extensive complementation tests against single mutant alleles and examination of cuticle phenotypes. For ATAC-seq experiments, patternless mutant mothers were trans-heterozygotes between two independent quintuple mutant recombinants, and one of the two mutant chromosomes also carried *His2Av-EGFP* to facilitate staging of embryos. The *opa* mutant allele (*opa$^{IIP32}$*) was obtained from the Schupbach-Wieschaus Stock Collection at Princeton University (*Jürgens et al., 1984*). For ATAC-seq experiments, *opa*/TM3, Sb were first crossed to *w; His2Av-EGFP* and embryos were collected from a cross between resultant *opa*/*His2Av-EGFP* adults. In the course of this study, we determined the nature of the *opa$^{IIP32}$* lesion as a single G > A point mutation (chr3R:4853998–4853998, dm6 assembly) that generates a missense allele converting a critical cysteine residue within the first predicted Opa C2H2 Zn-finger to tyrosine (C298Y). To determine the zygotic genotypes of the individual embryos produced from the cross between *opa*/*His2Av-EGFP* adults, single-nucleotide polymorphisms (SNPs) over the *opa* coding sequence were called using Samtools/BCFtools. Sequencing depth for these samples over variant bases within the *opa* locus were sufficiently deep (ranging from 47 to 104 independent reads per base) to allow for high-confidence *post hoc* genotyping. Whereas the wild-type chromosome was found to harbor SNPs that generate samesense mutations at a threonine and a proline residue (chr3R: 4868551 (G > T) and chr3R: 4868557 (T > C), respectively), a unique SNP (chr3R:4853998 (G > A)) yielding a missense mutation at cysteine 298 was found present in either 0,~50%, or 100% of base calls at that position for a given sample. Sample genotypes were therefore assigned according to the frequency of recovery of the presumed causative mutation on the *opa$^{IIP32}$* chromosome. Since this lesion is predicted to severely impact Opa DNA binding and mutant *opa$^{IIP32}$* cuticle phenotypes are indistinguishable from those produced from homozygous *opa$^{Df}$* (data not shown), we conclude the *IIP32* allele is effectively amorphic.

## CRISPR modification of the *opa* locus

CRISPR was performed by microinjection of *y sc v; nos >Cas9/CyO* embryos with ~60 pl of a mixture of plasmids encoding pU6-driven sgRNAs targeting *opa* and *ebony* (100 ng/ul each) and a plasmid for homology directed repair to insert either a 3x-myc tag or an EGFP-enhancer llama-tag (*Bothma et al., 2018*; *Kirchhofer et al., 2010*) at a concentration of 500 ng/ul. All injections were

performed in-house using a modified in-chorion approach originally developed by Nicolas Gompel and Sean Carroll (http://carroll.molbio.wisc.edu/methods/Miscellaneous/injection.pdf) which greatly facilitated high-throughput injection and enhanced survival of hundreds of embryos per day. The cited protocol was enhanced by beveling unclipped, freshly pulled injection needles at a 40° angle on a Narishige Needle Grinder (Narishige, Tokyo, Japan), resulting in reproducible needle preparations that easily pierced the chorion and minimized damage to injected embryos from irregularly surfaced tips. Targeting efficiency of *ebony* was used to select for likely 'jackpots' of CRISPR modified founders, as described in *Kane et al. (2017)*. The *opa* sgRNA target sequence was selected from the UCSC Genome Browser/CRISPOR pre-calculated sgRNA sequences (*Concordet and Haeussler, 2018*) and targets chr3R:4869148–4869167 (+ strand, dm6), yielding an expected Cas9-dependent double-strand break 4 bp downstream of the desired insertion site (immediately upstream and in-frame with the *opa* stop codon). The guide sequence oligonucleotides were synthesized with a leading G residue and terminal sequences compatible with insertion into BbsI-digested pU6-2xBbsI as described at http://flycrispr.org/protocols/grna/ (*Gratz et al., 2015*). Homology-directed repair (HDR) constructs were constructed using two 500 bp homology arms either terminating at the insertion site on the left or beginning at the expected Cas9 double-strand break on the right. Because the selected *opa* sgRNA target sequence spanned the site of insertion and would therefore be disrupted either in the HDR vector or in successfully targeted and repaired *opa* loci, we did not introduce sgRNA or PAM site mutations to prevent CRISPR targeting of the HDR plasmid or re-targeting of repaired loci. We constructed HDR sequences in a modified pBluescript KS vector where we introduced a *3xP3 > dsRed* eye marker into the vector backbone (pBS 3xP3) to facilitate negative selection of unwanted founders that incorporate the full HDR plasmid by homologous recombination. All DNA constructs for injection were prepared using a modified Qiagen EndoFree Midiprep protocol (Qiagen, Hilden, Germany), which enhanced survival of injected embryos compared with standard Qiagen Midiprep grade DNA preparations. Following injection, surviving embryos were raised to adulthood and single founders were crossed to five or six *w; Dr e/TM6c, Sb* flies. F1 progeny were scored for the proportion of *ebony* and vials with >50% *ebony* progeny were scored as 'jackpots' in which we induced greater-than-monoallelic targeting of the *ebony* locus. Up to six individual jackpot males from up to six jackpot founders were subsequently crossed to *w; Dr e/TM6c, Sb* females and stocks were established from F2 progeny. We favored jackpots that produced 85–100% *ebony* progeny. We observed a 32% survival rate of injected embryos (effectively 64% survival taking into account homozygous lethality of either the second-chromosomal *nos >Cas9* transgene or *CyO*), of these 40–60% of founders produced at least one *ebony* progeny, and 15–25% of surviving founders, overall, produced jackpot lines. Following selection against whole-pasmid integrants (*dsRed+* individuals), approximately 75% of selected F2 jackpot individuals were positive for the desired insertion. No difference in insertion efficiency was observed between *3xMyc* and the *EGFP-llama* cassettes. Positive insertions were identified by PCR and reared to homozygosity before PCR amplification of the modified locus (using primers targeting sites outside the original left and right homology arms) and sequence confirmation by Sanger sequencing. Between two and four stocks representing identical independently generated alleles were kept and confirmed for general stock viability and overall expression patterns/functionality of the modified locus, but core experiments were performed on single representative stocks isogenic for individual engineered chromosomes.

## Time-lapse imaging of *opa-llama* expression

Embryos produced from a cross between *w His2Av-RFP; bcd_{promoter} >EGFP/+* females and *w; opa-llama* males are maternally loaded with RFP-marked histones, low levels of free uniformly expressed EGFP, and are heterozygous for a llama-tagged *opa* allele. Embryos were dechorionated for 1 min in 4% Clorox (50% dilution of a commercially available 'concentrated' Clorox Bleach solution), washed extensively in dH$_2$O, and mounted in halocarbon oil sandwiched between a glass coverslip and a gas-permeable membrane mounted on an acrylic slide. Syncytial blastoderm embryos were identified on the basis of morphology and nuclear density, and time-lapse images were collected on a Leica SP8 WLL confocal microscope (Leica Microsystems, Wetzlar, Germany) with a 40 × 1.3 NA oil-immersion objective, using excitation wavelengths of 489 nm for EGFP and 585 nm for RFP. Images were captured at 512 × 240 pixels with 692 nm x-y pixel size (0.83x Zoom) and a 1 μm z-step spanning a z-depth of 12 μm, 115 μm pinhole size, and 110 Hz scan rate with bidirectional scanning. Series were collected at 30 s per xyz stack. We required that a movie begin within NC12

and for data collection to continue until at least NC14 + 72 min. This ensured not only that we quantified *opa* expression from the beginning of NC14, but also controlled for variation in developmental rates by allowing us to measure the duration of NC13. We required that NC13 would last no longer than 22 min and no shorter than 17 min for imaging to proceed. Fluorescence intensity was quantified by segmenting nuclei in the RFP channel and quantifying average, per nucleus intensity in the GFP channel. Nuclei outside of the *opa* expression domain were manually identified to estimate fluorescence background and plotted expression levels were calculated by subtracting the background estimate from the average intensity of nuclei within the observed *opa* expression domain. Values from three biological replicates (three individual embryos imaged on 3 different days from two independent crosses) were averaged to yield the plotted data.

Construction of *tub >opa* for maternal misexpression *opa* genomic sequences were amplified from a single squashed homozygous adult *opa-3xmyc* fly by standard PCR. Two PCR amplicons containing the entire *opa* coding sequence, omitting the large 14.1 kb first intron but retaining the small 140 bp second intron, were joined in-frame and inserted between the maternal *alpha-tubulin 67* c promoter and *sqh* 3'UTR in the transgenic vector pBABR-tub-MCS-sqh (*Hannon et al., 2017*). This yields maternal *alpha-tubulin* driven *opa-3xmyc* uniformly across the AP axis. Transgenic lines were generated by injecting 60 pl of a 200 ng/ul solution of the transgenic vector into *y sc v nos > phiC31 int; attP40*, outcrossing founders to *w; Sco/CyO* and selecting for mini-white positive transformants. Stocks were maintained unbalanced because of apparent synthetic dominant maternal effect lethality when maintained in trans to CyO that was not evident when stocks were maintained over a wild-type second chromosome.

## ATAC-seq

ATAC-seq was performed essentially as described in *Blythe and Wieschaus (2016)*. Briefly, embryos were staged at NC14 to the minute by observation of anaphase of the prior cell cycle (t = 0') and aging embryos to the desired stage. Embryos were maintained at constant room temperature to reduce variability in staging. Three minutes before the desired stage, a single embryo was dechorionated and macerated in Lysis Buffer (*Buenrostro et al., 2013*). Nuclei were pelleted by gentle centrifugation at 4°C at 500 RCF and the supernatant was discarded prior to freezing on dry ice. We have determined that one critical parameter in this process is the use of low-retention 1.5 ml microcentrifuge tubes (e.g. Eppendorf DNA LoBind tubes, manufacturer part number 022431021) (Eppendorf, Hamburg, Germany). Samples were briefly thawed on ice before tagmentation in a 10 µl solution as described previously. A second critical parameter was to ensure sufficient resuspension of the nuclear pellet during the addition of the tagmentation solution, which we achieve by extensive up-and-down pipetting upon addition of the solution. Samples were incubated for 30 min at 37°C with 800 RPM agitation in an Eppendorf Thermomixer HotShakey device. Following tagmentation, samples were cleaned up with the Qiagen MinElute Reaction Clean Up kit, eluting in 10 µl.

ATAC libraries were amplified using a set of modified Buenrostro primers that introduced Unique Dual Indexes (UDI). Buenrostro primers differ slightly from Illumina Nextera primers with the addition of 7 bp (Nextera Index Read 2 Primer/Universal or i5) or 6 bp (Nextera Index Read 1 Primer/i7) additional base pairs to the 3' ends of standard Nextera primer sequences (*Buenrostro et al., 2013*) (Illumina Inc, San Diego, California). Although we have not confirmed this directly, the additional bases in the Buenrostro primer design may yield improved amplification efficiency from ATAC libraries than standard Nextera primers as has been reported for a different Tn5 tagmentation-dependent assay, Cut and Tag (*Kaya-Okur et al., 2019*), see https://www.protocols.io/view/bench-top-cut-amp-tag-z6hf9b6). Therefore, we designed modifications of the original Buenrostro primer design rather than purchase commercially available UDI Nextera primer sets. The generalized UDI ATAC primer sequences are:

ATAC UDI Index Read 2 (i5):
5'- AATGATACGGCGACCACCGAGATCTACACnnnnnnnnTCGTCGGCAGCGTCAGATGT*G −3'
ATAC UDI Index Read 1 (i7):
5'- CAAGCAGAAGACGGCATACGAGATnnnnnnnnGTCTCGTGGGCTCGGAGATG*T −3'

Where the ATAC UDI Index Read two primer corresponds to the prior Buenrostro Universal primer that has been modified to introduce an i5 index sequence $(n)_8$ at the appropriate location based on Illumina primer designs. The ATAC UDI Index Read one primer corresponds to the indexed

Buenrostro primer designs, although we did not retain the same i7 barcodes. Because not all index combinations are compatible with one another, we used index pair combinations reported for the New England Biolabs NEBNext Multiplex Oligos for Illumina (96 Unique Dual Index Pairs) set (catalog number E6440, New England Biolabs, Beverly, Massachusetts), replacing the $(n)_8$ sequences above with the appropriate i5 and i7 barcode. Primers were synthesized with a terminal phosphorothioate bond (*) by IDT (Integrated DNA Technologies, Coralville, Iowa). No major differences were observed with library amplification or sequencing efficiencies between the UDI primer sets and the original Buenrostro designs.

Amplified libraries were checked on an Agilent Bioanalyzer (Agilent Biotechnologies, Santa Clara, California) to confirm the presence of distinct open and nucleosome-associated fragment sizes. Twelve embryos total were processed for the comparison between wild-type and 'patternless' mutant embryos: three embryo biological replicates per timepoint (NC14+12' or +72') and per genotype. For ATAC on *opa* mutant embryos, 22 embryos were collected from a cross between *opa/+* heterozygous adults and sequenced in two separate library pools. This yielded five wild-type embryos, three *opa* mutant embryos, and 14 heterozygous embryos. For ATAC-seq on NC13+12' wild-type and *tub >opa/+* embryos, six embryos were collected per genotype. Sample sizes were chosen by the following rationale: in prior studies, we have observed that three independent biological samples provide sufficient information to estimate the reproducibility of an observed result. In the case of the *opa* mutant ATAC experiment, we sequenced samples until at least three of each expected zygotic genotype was recovered. In the *tub >opa* experiment, we sequenced more than three samples simply because we had the choice of whether to sequence three replicates each at twice the depth, or sequence twice as many samples in order to fill the capacity of a sequencing lane. On the basis of Picard MarkDuplicates output (data not shown) increasing sequencing depth would not have provided additional information and would have only increased the duplication rate of the samples, so we chose to sequence more samples. Throughout the study, the term 'replicate' refers strictly to a biological replicate as samples were generated from individual embryos, that is, distinct biological samples.

## ChIP-seq

Per replicate, 200 cellular blastoderm stage $w^{1118}$ or *opa-3xmyc* embryos were collected, crosslinked and ChIPped as described in *Blythe and Wieschaus (2015)* for a total of three independent biological replicates. The ChIP antibody was an anti-myc tag polyclonal antibody from Sigma-Aldrich (C3956, Sigma-Aldrich, St. Louis, Missouri). ChIP-seq libraries were prepared using the NEBNext Ultra II library prep kit with the NEB Unique Dual Index primer set kit (New England Biolabs). Samples were sequenced at the Northwestern University NuSeq Core Facility on an Illumina HiSeq 4000 on single-end 50 bp mode. Demultiplexed reads were run through TrimGalore! (https://github.com/FelixKrueger/TrimGalore) and mapped to the dm6 assembly of the *Drosophila* genome using Bowtie2 with options `-fivePrimeTrim 5 N 0 -local` (*Langmead and Salzberg, 2012*). Suspected optical and PCR duplicates were marked by Picard MarkDuplicates (http://broadinstitute.github.io/picard/). To determine a set of high-confidence peaks, we called peaks using relaxed stringency with MACS2 (*Zhang et al., 2008*) followed by determination of reproducible peak regions per replicate using the Irreproducible Discovery Rate algorithm (*Landt et al., 2012*). First, we used Bedtools bamtobed (https://bedtools.readthedocs.io/en/latest/) to convert. bam formatted mapped, filtered primary sequencing reads to. bed format, which facilitated generation of pseudoreplicate datasets. Pseudoreplicates were generated by randomly splitting each replicate sample into two bed files. Overall pseudoreplicates were generated by pooling all replicates for either *w* or *opa-3xmyc* and randomly splitting reads into two separate bed files. Several peaks files were generated using MACS2 in preparation for IDR analysis. First, peaks for each individual *opa-3xmyc* sample (n = 3) were called against the pooled *w* sample data as a control. Having previously determined the MACS2 parameter *-d*, we bypassed model building (`-nomodel`) and manually specified the expected fragment size (`-extsize 149`). Relaxed conditions were specified by designating the option `-p 1e-3` as recommended by IDR. Samples were scaled to the larger dataset (`-scale-to large`). A second peaks list (n = 1) was generated for the pooled *opa-3xmyc* samples against the pooled *w* samples using the same MACS2 options. A third set of peaks (n = 6) were called for each pseudoreplicate for each biological replicate of *opa-3xmyc* against the pooled *w* samples, using the same MACS2 options. Finally, a fourth set of peaks (n = 2) were called for the two pseudoreplicates

of the pooled *opa-3xmyc* data against the pooled *w* data, using the same MACS2 parameters. We then performed IDR analysis on this set of peaks with threshold values of 0.02 for individual replicates and 0.01 for the pooled data sets. IDR was performed for pairwise comparisons between replicates (e.g. rep 1 vs rep 2, rep 2 vs rep 3, and rep 3 vs rep 1) using as a reference the peak list from pooled *opa-3xmyc* samples. IDR options were `-input-file-type narrowPeak -rank p.value -soft-idr-threshold 0.02` and `-use-best-multisummit-idr`. IDR was then subsequently performed on each pair of pseudoreplicates for each individual biological replicate (e.g. rep1 pseudo 1 vs rep 1 pseudo 2...) using the same IDR options. Finally, IDR was performed on the pooled sample pseudoreplicates using the same IDR options except `-soft-idr-threshold` was 0.01 instead of 0.02, per the recommendation of the IDR instructions. The largest number of reproducible peaks between replicates was 881, and so we took the top 881 peak regions (ranked by p-value) from the list of pooled peaks and filtered out two regions that mapped to non-canonical chromosomes, leaving 879 high confidence peaks. Consequently, this is a conservative estimate of true Opa binding sites but is determined by more rigorous criteria (reproducibility between biological replicates) than either arbitrary p-value thresholding or simply taking the top N peaks.

## ATAC-seq analysis

Demultiplexed reads were trimmed of adapters using TrimGalore! and mapped to the dm6 assembly of the *Drosophila* genome using Bowtie2 with option -X 2000. Suspected optical and PCR duplicates were marked by Picard MarkDuplicates. Mapped, trimmed, duplicate marked reads were imported into R using the GenomicAlignments (*Lawrence et al., 2013*) and Rsamtools libraries (http://bioconductor.org/packages/release/bioc/html/Rsamtools.html), filtering for properly paired, non-secondary, mapped reads, with map quality scores greater than or equal to 10. Reads with mapped length less than or equal to 100 bp were considered to have originated from 'open' chromatin.

For the *opa* mutant ATAC-seq experiment, embryos were collected blind to the zygotic genotype, but because experiments were performed on single embryos, genotypes could be determined post-hoc by calling SNPs over the *opa* locus using BCFtools (*Danecek et al., 2011*).

Peak regions for all accessible regions present between ZGA and gastrulation were called using MACS2 with options `-f BAMPE -q 1e-5 -nomodel` on a merged dataset comprising all 'open' ATAC reads from both wild-type and 'patternless' mutant embryos, and both NC14+12 and +72 timepoints.

DEseq2 (*Love et al., 2014*) analysis was performed by counting the number of 'open' ATAC seq reads overlapping each peak identified by MACS2. For the comparison between wild-type and 'patternless' mutant embryos over NC14+12 and +72, the design parameter passed to DESeq2 was ~genotype + time + genotype:time. For other comparisons (*opa* gain or loss of function at single timepoints) the design was ~genotype only. Regions with an adjusted p-value of less than 0.01 and absolute $\log_2$ fold-changes > 1 were considered to be statistically significant in the relevant test.

To determine the different dynamic peak classes (*Figure 4*), we first undertook a clustering based approach to explore how many different classes we could identify within the dataset. To do this, we first averaged the number of reads per peak region for each sample and then scaled the data for each peak region by dividing the mean reads for a peak region by the sum of all mean reads for a peak region (i.e. so that the sum of the scaled reads for a peak region, $\text{sum}_{region}$(wt 12, wt, 72, mut 12, mut 72), would equal 1). Next, we performed k-means clustering with a variable number of cluster centers, and plotted the average per-cluster profile to visualize average behavior of clustered regions. We found that 10 was the minimum number of cluster centers that would capture all the unique patterns present in the data, that fewer clusters would combine similar but qualitatively different classes, and more clusters would subdivide clusters into relatively similar subsets. We note that over the two-year course of this study, attempts were made to replicate this analysis on three different Apple (Apple Computer Inc, Cupertino, CA) computers running different base operating systems, versions of R, and versions of dependent libraries. For reasons that are not clear, clustering based approaches were not strictly reproducible across differently configured computers despite identical input datasets and identical scripting of the analysis code. To be clear, similar cluster types and minimum cluster number were called across different systems; however, the numeric order of the clusters, as well as the number of peaks assigned to each cluster would vary between systems. Because of this, we could not rely on clustering alone to reproducibly describe our results. Therefore, we took an alternative approach to categorizing peak classes that depended on the statistical

output of DESeq2, whose output was identical across different computer systems. We paired statistical tests from DESeq2 to define classes. For instance, to define regions that were uniformly open early and uniformly lost accessibility by gastrulation, we required both wild-type and mutant samples to have a statistically significant difference across timepoints, with a log2 fold change of −1 or less. On the basis of these paired DESeq2 criteria, we reproduced each of the 10 peak class types predicted by clustering. We note that the final number of categorized peaks (2917) is substantially lower than the number of peaks that score above the significance + log2 fold change thresholds of 0.01 + |1| (6775). This is due to the fact that we require pairs of DESeq2 tests to score above significance thresholds for assignment to a class. The remaining 3858 regions only score as significantly different in one of the four tests. The details of this analysis are provided in a markdown document (*Source code 1*) and all processed data as well as a user-executable version of the markdown document are provided at the lab's GitHub Page: https://github.com/sblythe/Patternless_ATAC (*Blythe, 2019*; copy archived at https://github.com/elifesciences-publications/Patternless_ATAC).

## Generating reporter constructs based on ATAC-seq data

We used ATAC-seq coverage to delineate the sequence to test for potential enhancer activities associated with *wg-2.5*, *wg-1*, *odd-late*, *slp1 '5'*, *18 w 1* and *18 w 2*. In general, coverage at peak regions was plotted on the UCSC genome browser and views were zoomed out to identify flanking regions of low accessibility. We hypothesized that functional genomic elements would be defined by extended regions of accessible chromatin flanked by inactive, low-accessibility regions. Primer pairs were designed to amplify peak regions plus a small amount of flanking 'low-accessibility' DNA and were cloned into a Gateway entry vector pENTR (Thermo Fisher Scientific, Waltham Massachusetts). The exception to this overall strategy was in the case of the *wg-2.5* and −1 regions which are adjacent to one another (*Figure 3A*). in which case we delineated the two regions by the midpoint between the two major peaks of chromatin accessibility. Once we had generated one pENTR-enhancer clone by TA-cloning of a PCR product, it was no longer necessary, efficient, or desirable to perform TA cloning. Subsequent pENTR clones were made by excising the original insert and replacing with new candidate enhancer fragments via Gibson Assembly (NEB HiFi Assembly Kit, New England Biolabs). The candidate enhancer fragments were shuttled to the transgenic vector pBPGUw (*Pfeiffer et al., 2008*) upstream of a minimal synthetic promoter sequence driving Gal4 using standard Gateway cloning techniques (LR Clonase, Thermo Fisher Scientific). Transgenic lines were established as described above by insertion into the *attP40* landing site.

The genomic coordinates corresponding to regulatory elements tested in this study are: *odd-late*: chr2L:3602087–3603534; *slp1 '5' variant*: chr2L:3817360–3818884; *wg −1*: chr2L:7305311–7306265; *wg −2.5*: chr2L:7303832–7305569; *18 w 1*: chr2R:20091666–20092774; *18 w 2*: chr2R:20071902–20073306.

## In situ hybridizations

Probe sequences for in situ hybridizations were generated by PCR against cDNA clones for the target transcript. Reverse primers included a leading T7 promoter sequence to facilitate probe synthesis by in vitro transcription using an Ambion MegaScript T7 kit (Thermo Fisher Scientific) supplemented with digoxigenin-11-UTP (Roche, Sigma-Aldrich) following standard procedures. In situ hybridizations were performed according to standard procedures using a hybridization temperature of 65°C, which was found to significantly reduce background compared with 56°C. For reducing bias in the evaluation of gene expression in 'patternless' mutant embryos, color development reactions were performed side-by-side with a wild-type control. Color development was evaluated by observing only the wild-type sample and stopping both reactions once it was determined that wild-type samples had just reached optimal signal. Staining levels in mutant embryos were then evaluated following termination of the color development reaction. For evaluating sensitivity of a reporter to *opa* loss of function, we performed in situ hybridizations against *gal4* on two samples in parallel, a wild-type sample of *test enhancer >gal4* homozygous males crossed to $w^{1118}$ females, and embryos from a cross between *opa$^{IIP32}$/TM3, Sb twi >gal4 UAS >GFP* females and *w; (test enhancer >gal4); opa$^{IIP32}$/TM3, Sb twi >gal4 UAS >GFP* males, which allowed us to distinguish test enhancer >reporter expression in *opa* homozygous mutant embryos by their lack of *twi >gal4*

staining in the ventral mesoderm. Color development was monitored in the wild-type sample, and mutant embryos were ultimately hand selected from the *twi >gal4* marked sample, after ensuring that wild-type expression patterns matched the *twi >gal4*-positive expression patterns, save for the broad domain of *twi >gal4* expression. Images shown in figure panels are of similarly staged wild-type samples and *twi >gal4* negative (*opa* mutant) samples imaged under identical conditions. T7-antisense Dig-UTP riboprobe template-generating primer sequences are as follows:

> *gal4-F*: 5'- TGCGATATTTGCCGACTTA −3'
> *gal4-R*: 5'- TAATACGACTCACTATAGGGAACATCCCTGTAGTGATTCCA −3'
> *ems-F*: 5'- AGCATCGAGTCCATTGTGGG −3'
> *ems-R*: 5'- TAATACGACTCACTATAGGGTATCTCCTGGCCGCTTCTCT −3'
> *Kr-F*: 5'- CTGGATGTGTCGGTGTCTCC −3'
> *Kr-R*: 5'- TAATACGACTCACTATAGGGCGGTAACAAGCATACGGT −3'
> *byn-F*: 5'- ACGAACCACGTGTGCATCT −3'
> *byn-R*: 5'- TAATACGACTCACTATAGGGTAGGAACTGCTGAAGCAACCA −3'
> *ush-F*: 5'- GAAGGTGCGAGTGAAGTGGA −3'
> *ush-R*: 5'- TAATACGACTCACTATAGGGCGAATGGGCTTGTTTTT −3'
> *sog-F*: 5'- CCATTGTGTGTGCCAGTGTG −3'
> *sog-R*: 5'- TAATACGACTCACTATAGGGCACTTGGTCTCTCCGTTCA −3'
> *sna-F*: 5'- CCGGAACCGAAACGTGACTA −3'
> *sna-R*: 5'- TAATACGACTCACTATAGGGCCAGCGGAATGTGAGTTTGC −3'

## Acknowledgements

We gratefully acknowledge Eric Wieschaus, in whose laboratory this project was initiated, for his generosity, support, and encouragement. We also thank Angelike Stathopoulos, Erik Clark, Melissa Harrison, Urs Schmidt-Ott, Judy Kassis, Jeff Farrell, Hernan Garcia, Jacques Bothma, Eileen Furlong, Nicolas Gompel, Dan McKay, Carole LaBonne, Rich Carthew, Erik Andersen, and Greg Beitel for helpful conversations. The authors gratefully acknowledge the services provided by the NUSeq Core facility and the Biomedical Imaging Facility at Northwestern University, as well as the Genomics Core Facility of the Lewis-Sigler Institute at Princeton University. This work was supported by a Searle Leadership Fund for Life Sciences Award and Northwestern University Startup Funds to SAB.

## Additional information

### Funding

| Funder | Grant reference number | Author |
| --- | --- | --- |
| Searle Leadership Fund for Life Sciences | New Faculty Start Up Funds | Isabella V Soluri<br>Lauren M Zumerling<br>Omar A Payan Parra<br>Eleanor G Clark<br>Shelby A Blythe |
| Northwestern University | New Faculty Start Up Funds | Isabella V Soluri<br>Lauren M Zumerling<br>Omar A Payan Parra<br>Eleanor G Clark<br>Shelby A Blythe |

The funders had no role in study design, data collection and interpretation, or the decision to submit the work for publication.

### Author contributions

Isabella V Soluri, Formal analysis, Investigation, Visualization; Lauren M Zumerling, Investigation, Visualization; Omar A Payan Parra, Investigation, Writing - review and editing; Eleanor G Clark, Investigation, Visualization, Writing - review and editing; Shelby A Blythe, Conceptualization, Resources, Data curation, Formal analysis, Supervision, Funding acquisition, Investigation, Visualization, Methodology, Writing - original draft, Project administration, Writing - review and editing

**Author ORCIDs**
Omar A Payan Parra ⬤ http://orcid.org/0000-0002-0005-8355
Shelby A Blythe ⬤ https://orcid.org/0000-0003-4986-2579

**Decision letter and Author response**
Decision letter https://doi.org/10.7554/eLife.53916.sa1
Author response https://doi.org/10.7554/eLife.53916.sa2

## Additional files

**Supplementary files**
• Source code 1. DESeq analysis markdown. This file is a markdown document that shows how critical aspects of the analysis described in this paper was performed. It is intended to provide transparency on how this was done from both a scientific and a methodological standpoint. We note that both this document, as well as all input data files necessary for reproducing this analysis are provided on the lab's github page https://github.com/sblythe/Patternless_ATAC.

• Supplementary file 1. Curated segmentation network CRMs. CRMs with activity up to early germ-band extension stage and associated with segmentation network genes were obtained from the Redfly database. Some manual curation of this list was performed: large *engrailed* CRMs corresponding to two spatially separate ATAC regions (*Supplementary file 2*) were split and named accordingly (e.g. *en_H* was split into *en_H1* and *en_H2*). In addition, a new CRM corresponding to the *odd* late interstripe enhancer (see *Figure 6*) was added to this listing.

• Supplementary file 2. Annotated ATAC-seq peaks list. This file contains a tab-delimited table with 26328 rows (peaks) and 48 columns, plus one header row and one column of row numbers. In addition, descriptions of each column or set of columns are provided at the head of the document and are demarcated by the comment character [#]. In summary, the table contains the genomic locations of all identified ATAC peaks, plus outputs from the DESeq2 analysis for differential enrichment for all the comparisons described in the text. Groupings of differentially accessible sites (as described in *Figure 4* and supplements) are represented in this table as logical vectors indicating membership in a group. Logical vectors are also provided to allow for cross-referencing with overlapping regions in the Opa ChIP-seq peaks list, as well as generic genomic features (intron, exon, et c.). Finally, a column with the number of Opa motifs per peak is provided.

• Supplementary file 3. Annotated Opa ChIP-seq peaks list. This file contains a tab-delimited table with 879 rows (peaks) and 19 columns, plus one header row and one column of row numbers. In addition, descriptions of each column or set of columns are provided at the head of the document and are demarcated by the comment character [#]. In summary, the table contains the genomic locations of all IDR filtered Opa ChIP-seq peaks, plus outputs from DESeq2 analysis for differential accessibility as measured by ATAC-seq under conditions of loss- and gain- of function of Opa. The final column indicates the number of Opa motifs within each ChIP peak region.

• Transparent reporting form

**Data availability**
Data have been deposited in the Gene Expression Omnibus with accession number GSE141538. Processed datasets including a step-by-step demonstration of the data analysis process have been made available on GitHub: https://github.com/sblythe/Patternless_ATAC (copy archived at https://github.com/elifesciences-publications/Patternless_ATAC). Expression patterns of Vienna Tiles are from https://enhancers.starklab.org.

The following dataset was generated:

| Author(s) | Year | Dataset title | Dataset URL | Database and Identifier |
|---|---|---|---|---|
| Blythe SA | 2020 | Zygotic pioneer factor activity of Odd-paired/Zic is necessary for establishing the Drosophila Segmentation Network | https://www.ncbi.nlm.nih.gov/geo/query/acc.cgi?acc=GSE141538 | NCBI Gene Expression Omnibus, GSE141538 |

The following previously published datasets were used:

| Author(s) | Year | Dataset title | Dataset URL | Database and Identifier |
|---|---|---|---|---|
| Hannon CE, Blythe SA, Wieschaus EF | 2017 | Data from: Concentration dependent binding states of the Bicoid Homeodomain Protein | https://www.ncbi.nlm.nih.gov/geo/query/acc.cgi?acc=GSE86966 | NCBI Gene Expression Omnibus, GSE86966 |
| Blythe SA, Wieschaus EF | 2015 | Data from: ChIPseq measurement of transcriptional engagement and replication stress at the Drosophila Mid-blastula Transition | https://www.ncbi.nlm.nih.gov/geo/query/acc.cgi?acc= GSE62925 | NCBI Gene Expression Omnibus, GSE62925 |
| Harrison MM, Li X, Kaplan T, Botchan MR, Eisen MB | 2011 | Data from: Zelda binding in the early Drosophila melanogaster embryo marks regions subsequently activated at the maternal-to-zygotic transition | https://www.ncbi.nlm.nih.gov/geo/query/acc.cgi?acc=GSE30757 | NCBI Gene Expression Omnibus, GSE30757 |

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
