## [Decision Letter]

Thank you for submitting your article "Zygotic pioneer factor activity of Odd-paired/Zic is necessary for establishing the *Drosophila* Segmentation Network" for consideration by *eLife*. Your article has been reviewed by three peer reviewers, and the evaluation has been overseen by Oliver Hobert as the Reviewing Editor and Kevin Struhl as the Senior Editor. The following individual involved in review of your submission has agreed to reveal their identity: Erik Clark (Reviewer #2). The reviewers have discussed the reviews with one another and the Reviewing Editor has drafted this decision to help you prepare a revised submission.

All reviewers are in agreement that this is a very nice, interesting study that is of general importance. No additional experiments are required but we all agree that there are a number of editorial changes, clarifications and writing changes that need to be implemented before the manuscript becomes acceptable. Those are detailed in the reviews appended below. In addition, reviewer #1 suggested in points #1 and #2 an additional analysis of the ChIP peaks, which require no further experimentation, but may provide some additional insights; please try to perform this analysis (the experimental analysis suggested in point #3 by reviewer #1 was deemed to not be essential and does not need to be done).

Reviewer #1:

Soluri et al. identified widespread changes in accessibility of cis-regulatory elements of patterning genes as the *Drosophila* embryo undergoes gastrulation. Using a powerful quadruple mutant embryo and ATAC-seq, the authors demonstrate that the majority of these changes in accessibility are driven by developmental time and not dependent on patterning itself. They identify Odd-paired (Opa) motifs underlying regions that gain accessibility during gastrulation and using a combination of ChIP-seq and ATAC-seq demonstrate that Opa has some defining features of pioneer factors. In general, this is a well-executed paper addressing an important and interesting developmental question.

1) It would be useful to compare the number of peaks identified without using the IDR analysis for comparison with other data sets that do not use a tagged allele and therefore cannot use an IP from wild type as control. If additional peaks are identified in this less stringent analysis are these peaks enriched for a centered Opa motif? This might suggest that the stringent methods used are only identifying the most robust Opa-binding sites. Given that the ATAC-seq analysis on the opa mutant is only focused on these regions some additional information could be gained from looking at potential additional sites.

2) Expanding the ChIP analysis is especially important given that the ATAC-seq analysis is centered only on those stringently identified Opa ChIP peaks. It would be important to more globally discuss the changes in accessibility identified in the opa mutant. These data appears to be all included in the comprehensive ATAC-seq peak list, but should be discussed. How many loci overall gain and lose accessibility? How many of these overlap the stringently called Opa peaks? The focus on stringently called Opa peaks is understandable, but to more broadly assess the impact of the loss of Opa on accessibility, it is important to report all identified changes and not just those that overlap direct targets. Are there motifs selectively enriched in the Opa-dependent vs. Opa-independent regions bound by Opa that might suggest factors that maintain accessibility in the absence of Opa? Is there an enrichment for any genomic regions (enhancers, promoters)?

3) While it is nicely demonstrated that Opa influences chromatin accessibility at sites where it is bound, the downstream effects of this accessibility remain largely unanalyzed. It could be useful to have some orthogonal analysis of the role of Opa on gene expression. Mutating the Opa-binding sites in the odd-late reporter and seeing if that is not expressed would be a useful demonstration of the direct role of Opa at this CRM. Alternatively, the global effect of Opa-mediated accessibility on gene expression could be analyzed by RNA-sequencing of either the opa mutant or the tub-opa overexpression.

4) The impact of this manuscript would be strengthened if the Introduction and Discussion were streamlined for clarity.

Reviewer #3:

Soluri et al. address the important question of the dynamic acquisition of genome accessibility during key developmental transitions. Specifically they focus on the 1h period of *Drosophila* embryogenesis, from ZGA to gastrulation, during which regulatory networks establish segmental identities across the A/P axis.

By performing single embryo ATAC-seq on carefully staged *Drosophila* embryos prior to and at gastrulation, they identify a novel set of cis-regulatory elements that gain accessibility at later stages. Through an elegant genetic approach, where all A/P maternal cues are depleted to generate embryos with a uniform unique lineage, the authors were able to distinguish cis-regulatory elements that gain accessibility dynamically, specifically at late stages but in a patterning-dependent fashion.

The analysis of the sequences enriched in this category of ATAC-seq peaks revealed the over-representation of opa motif. While its role as a patterning gene during segmentation was already known, opa's requirement for genome accessibility is novel and nicely demonstrated by both loss and gain of function approaches in this manuscript. Overall I find the experiments presented in this work well-designed and the data of high quality. The manuscript is well written, and I particularly appreciated the tone of the last part of the Discussion where the limits of the study are well-stated.

General comments:

The WT and quintuple mutant embryos are compared at exact similar timing. Can the author comment on the developmental timing status of the quintuple mutant in terms of developmental delays?

Since a novel exciting finding of this study is the fact that opa acts as pioneer factor, it would be exciting to discuss a) the defining properties of a pioneer factor, and b) which of these properties opa seems to fulfil and which ones are still unclear. In such a discussion, a comparison with Zelda, which is the other pioneer factor shown to act as a quantitative temporal timer of gene expression, would be interesting. Additionally, the evidence that opa is able to engage its targets even in the context of nucleosomal DNA is currently absent. I agree that the bioinformatic analysis performed by the authors is consistent with this hypothesis; however, to avoid future confusion, it would be ideal to state it clearly in the Discussion.

Could the authors provide statistics: how many enhancers are analysed in Figure 3C, D and in Figure 1A? In Figure 3E, the number of peaks is indicated, but as an enhancer can exhibit multiple peaks, it's difficult to infer the number of CRMs considered.

Specific comments:

– The manuscript starts by focusing on a small set of known CRM within the segmentation network. Could the authors give a number for the number of segmentation network enhancers for which they observe a dynamic change in chromatin accessibility? Would it be possible to extend this analysis to another type of enhancers, such as the DV patterning network?

– Since the ATAC-seq data have been performed in carefully staged single embryos, it would be interesting to extract more information than the small set of segmentation enhancers.

– Since the authors performed opa ChIP-seq, can they examine if there is a correlation between the number of opa binding sites and the timing of accessibility?

– Could the authors discuss similarities and differences with the pioneer factor Zelda, as several studies (Foo et al., 2014, Dufourt et al., 2018, Yamada et al., 2019) demonstrate its function as a quantitative timer?

– In Figure 6, the authors employ a gene expression analysis on a subset of opa-dependent CRM to support the conclusion that 'the primary function of opa is to modulate the temporally restricted accessibility of a subset of critical CRM'. Although not essential for this manuscript, it would be very exciting to build MS2 reporter transgenes for these critical CRM (for example odd-late) and examine the effect of adding extra opa sites to see if this is sufficient to elicit a premature expression (again like Zelda does, as shown in Yamada et al. and Dufourt et al.).

---

## [Author Response]

Reviewer #1:[…] 1) It would be useful to compare the number of peaks identified without using the IDR analysis for comparison with other data sets that do not use a tagged allele and therefore cannot use an IP from wild type as control. If additional peaks are identified in this less stringent analysis are these peaks enriched for a centered Opa motif? This might suggest that the stringent methods used are only identifying the most robust Opa-binding sites. Given that the ATAC-seq analysis on the opa mutant is only focused on these regions some additional information could be gained from looking at potential additional sites.

See below in the response to point 2 how we have incorporated a broader peaks list into the description of the effect of Opa on chromatin accessibility. We chose to perform IDR analysis in order to report the most high-confidence peaks from the biological triplicate samples used in our ChIP-seq analysis. This is done primarily to employ a rigorous, systematic, data-driven thresholding approach to declaring a set of peaks. We view this as a feature, not a drawback of our analysis. We cannot stress this enough: if someone wants to reproduce these studies, the approach we have taken here is the best method we are aware of to ensure that what we report here is not only true in our hands but will likely also be observed in independent hands. However, we also understand the reviewer’s point of view: are we throwing away any information by being rigorous? Yes: in the “non-IDR” peaks there are additional regions that bind Opa and that contain a centered Opa motif (numbers below). They however tend to be of lower overall intensity and show larger variance in intensities between replicates than we feel confident in including with the set of peaks we have concentrated on. In practical terms: these are not the parts of the genome where one would want to really focus on studying Opa function. We have provided the raw data, as well as summaries of coverage (.wig formatted files) to GEO for any investigators interested in reproducing these analyses and focusing on this broader spectrum of Opa binding sites.

The IDR approach is to call peaks with MACS2 using an extremely high (i.e., lax) p-value cutoff (0.05) and then to use the reproducibility between replicates of peak score ranks within this list to determine which peaks are robust, reproducible Opa targets. MACS is very good at reporting any miniscule significant fold difference between control and experimental ChIP samples, and invariably the least robust of these called peaks are either noise that fell on the wrong side of the p-value cutoff, or really weak, noisy interactions that may not warrant a ton of attention. For the sake of transparency in this analysis, we have now indicated in the description of the results based on Opa ChIP the number of peaks we started with (see below) and what was left after IDR filtering. We have also added a pair of columns to the ATAC peaks list that annotates whether an ATAC peak overlaps with the lax or the IDR filtered peaks list.

The lax MACS2 peaks list on all replicates pooled together that was used as input for IDR analysis contains 3553 peaks, of which 879 passed the IDR filter (24.7%). We counted up the total number of Opa motifs in the entire genome (n = 69373, 80% match to the Opa PWM as reported by MEME). Of these, 2462 (3.5%) overlap with the entire peak list. Within this set of Opa motifs in all peaks, 949 (38.5%) motifs are present in the IDR peak list, and the remaining 1513 (61.5%) are in the non-IDR peaks. Therefore, IDR selects 24.7% of peaks but these have 38.5% of the available motifs, suggesting that IDR enriches for peaks with some subset of available motifs. The number of Opa motifs per peak ranges from 0 to 9 in the IDR peaks, and similarly from 0 to 8 in the non-IDR peaks. However, the number of peaks with zero Opa motifs is substantially different between these two peak classes. While only 28.9% of IDR peaks have zero Opa motifs, 55.3% of non-IDR peaks have zero Opa motifs. Therefore, IDR selects for peaks that are more likely to contain at least one Opa motif. For all peaks that contain at least one Opa motif, there is a similar average distance between the peak summit and the nearest Opa motif of 17.24 ± 3.90 bp (IDR) and 21.21 ± 3.31 bp (non IDR).

We feel that we have a strong rationale for the choice to use IDR for deciding on a set of ChIP-seq peaks to focus on and have therefore not re-analyzed our data by expanding our peaks list. To address the reviewer’s more general point, we have (in the response to point 2 below) included analysis of Opa loss- and gain-of-function effects over the entire ATAC-seq peaks list. This directly addresses the point of broadening the analysis of our analysis of the effects of Opa on the system.

2) Expanding the ChIP analysis is especially important given that the ATAC-seq analysis is centered only on those stringently identified Opa ChIP peaks. It would be important to more globally discuss the changes in accessibility identified in the opa mutant. These data appears to be all included in the comprehensive ATAC-seq peak list, but should be discussed. How many loci overall gain and lose accessibility? How many of these overlap the stringently called Opa peaks? The focus on stringently called Opa peaks is understandable, but to more broadly assess the impact of the loss of Opa on accessibility, it is important to report all identified changes and not just those that overlap direct targets. Are there motifs selectively enriched in the Opa-dependent vs. Opa-independent regions bound by Opa that might suggest factors that maintain accessibility in the absence of Opa? Is there an enrichment for any genomic regions (enhancers, promoters)?

This is true: most of these data were included in the comprehensive ATAC peaks list but we have now added a paragraph to the text describing Figure 5 and Figure 5—figure supplement 2 that describes the overall effect of Opa vis a vis the complete set of peaks defined in the ATAC analysis. We have also added to the ATAC peak list columns containing logical vectors that allow for filtering on the basis of membership in the following categories: 1) intergenic; 2) promoter; 3) exon; 4) intron; 5) In ‘lax’ Opa ChIP-seq peaks list’ 6) In ‘IDR’ Opa ChIP-seq peaks list. The first four of these are for the purposes of asking about association of a peak with a particular genomic feature. The last two facilitate comparison between the ATAC peaks and the ChIP peaks and allow for the ‘broader’ analysis requested in point 1 above.

To summarize, we demonstrate that examination of the effect of Opa on accessibility in the broad ATAC peak list allows us to estimate the degree of indirect effects of opa on chromatin accessibility. We observe indirect opa-dependent gains and losses in accessibility by this analysis. None of the regions that gain accessibility overlap with any form of the Opa ChIP Peaks list, these are all indirect effects and there are no associated enriched motifs. ~44% of the regions that go down overlap with the stringent Opa ChIP peaks, and an additional ~20% of these regions are included in the lax Opa ChIP peaks. The remaining 36% of these regions are not represented in any form of the Opa ChIP peaks and have moderate enrichment for a long adenine-rich motif with no good match to the TF motif databases. (There is a poor match to a Zn-finger TF named *jim*, but *jim* is not expressed in early embryos, so this is not a good candidate factor). We feel this further supports the need to focus on direct targets of Opa in the analysis.

We now note that the regions that are directly regulated by Opa are enriched for intergenic regions and very strongly depleted for promoter regions, similar to the overall set of dynamic regions from the ATAC data (for symmetry, we have now also added this information to the text describing the overall ATAC dataset (Figure 4)). We have also incorporated Kvon/Stark’s Vienna Tile dataset of validated enhancers in both the ATAC and the Opa analysis to substantiate the functionality of ‘intergenic’ regions we have identified. Opa enriches for Vienna Tiles with the functional characterization of “PairRule”, which fits nicely with expectations. We note here, however, that the Vienna Tile dataset is incomplete in that it only represents about 14% of the noncoding genome. It is therefore good for finding enriched enhancer classes but is likely rife with false negatives. We have carefully worded our interpretation of the enrichment analysis of Vienna Tiles to ensure we aren’t making conclusions based on the lack of enrichment for a particular class.

3) While it is nicely demonstrated that Opa influences chromatin accessibility at sites where it is bound, the downstream effects of this accessibility remain largely unanalyzed. It could be useful to have some orthogonal analysis of the role of Opa on gene expression. Mutating the Opa-binding sites in the odd-late reporter and seeing if that is not expressed would be a useful demonstration of the direct role of Opa at this CRM. Alternatively, the global effect of Opa-mediated accessibility on gene expression could be analyzed by RNA-sequencing of either the opa mutant or the tub-opa overexpression.

Thank you. We are currently performing orthogonal analysis of the role of Opa on gene expression as part of several follow-up studies to this work which will be reported separately.

4) The impact of this manuscript would be strengthened if the Introduction and Discussion were streamlined for clarity.

We have edited the Introduction and Discussion (primarily the Discussion) to streamline for clarity.

Reviewer #3:[…] General comments:The WT and quintuple mutant embryos are compared at exact similar timing. Can the author comment on the developmental timing status of the quintuple mutant in terms of developmental delays?

We comment here in depth and include a summary note in the manuscript.

Depending on how you want to think about it, this is a complicated question, namely, “What controls the timing of development, especially when you eliminate (or flatten) embryonic patterning cues.” And rather than just hand-wave here, we wanted to try to get some numbers. Without going too far off the philosophical deep end, we reasoned that the early timepoint should be synchronous given that wild-type and “patternless” mutants both complete the maternally-controlled phase of development with the same number of cell divisions, which themselves proceed with normal cell cycle periods, at least NC11-13 which we can easily observe. Thereafter, however, many of the zygotic ‘mileposts’ whereby the passage of time could be measured are themselves affected by the mutant phenotype. One patterning-independent marker of time is the process of cellularization, and this process completes at a morphological level in “patternless” mutants with indistinguishable kinetics from wild-type. We note that the major zygotic cellularization genes are all included in the set of regions that lose accessibility uniformly in the late timepoint (i.e., this process is equivalent between genotypes).

After cellularization, a wild-type embryo gastrulates, undergoing tremendous morphological changes over a 15-20 minute period. In contrast, the “patternless” mutants remain as an epithelial monolayer for approximately 9-10 hours before initiating some kind of morphological transformation that we have not yet examined carefully. Wild-type and mutant embryos therefore diverge morphologically at the moment of gastrulation. However, during gastrulation in a wild-type embryo, zygotic mitoses resume in distinct mitotic domains that are themselves clock-like in that they always occur in the same order and in the same place, and that the timing of these domains is linked to spatially restricted patterning events (Foe V.E., Development 1989; Edgar B.A. and O’Farrell, P.H., Cell 1989; Momen-Roknabadi, A., et al., Cell Reports2016). We noticed in the course of these experiments that although “patternless” mutants remain as an epithelial monolayer, they continue to perform metasynchronous mitotic divisions albeit over much longer periods than the syncytial divisions. We do not understand the mechanistic basis for these additional divisions in the mutant, but they may represent continual adherence to the biological timing mechanisms that dictate the precise timing of zygotic mitotic domains. Based on V. Foe’s mitotic domain maps, and our limited expression profiling of mutants (e.g., Figure 2), we predicted that the mutant embryos would be fated to contribute to mitotic domain 13, which should initiate mitosis by 86 minutes after the start of NC14 (albeit at 25°C). We live-imaged 5 “patternless” mutant embryos expressing Histone RFP under the confocal at an approximate temperature of 21°C. Of these, three initiated mitosis before NC14+120’ (range 92’-119’). We lost patience around 120’ with the other two embryos, which did not divide by this time. This suggests that by 120 minutes post NC14, some temporal heterogeneity has arisen in the mutant embryos, with some embryos sticking somewhat closely to the expected zygotic mitotic clock, albeit with a broader variance (>>20’) than would be expected in a wild-type sample.

To generate an additional estimator of synchrony between wild-type and “patternless” mutant embryos, we imaged the onset of posterior *wingless* expression using a *wg-MS2* CRISPR allele (kindly provided by Hernan Garcia, Cal Berkeley). We measure the onset of posterior *wg* expression at 36.0 ± 1.73 minutes in wild-type, and at 36.7 ± 1.89 minutes in “patternless” mutants (n = 3 per genotype, *p* = 0.67, t-test). If we instead score the timepoint at which the posterior *wg* expression domain is fully ‘on’, this occurs at 42.3 ± 2.25 minutes in wild-type, and 46.17 ± 6.33 minutes in mutants (n = 3 per genotype, *p =* 0.57, t-test). These measurements suggest that at earlier timepoints corresponding to the period in which we performed our ATAC experiments, timing between the two genotypes is more synchronous.

Finally, we point out that our NC14+72’ timepoint replicates have similarly clustered distributions within the PCA analysis suggesting that we have measured a reproducible, similarly variant event in each genotype. We also do identify a substantial number of genomic regions that undergo changes in accessibility (gains or losses) in both genotypes, and that motif enrichment supports the idea that these regions are controlled by uniformly-expressed factors. We feel that although the broader question is interesting of whether the passage of time in a multiply mutant embryo is the same as in wild type, this one-hour period in which we have chosen to perform our measurements remains synchronous enough between the two genotypes that we are confident in our delineation of patterning-dependent and -independent events.

A study of this broader question of timing in the absence of patterning inputs will be performed elsewhere. However, we have added a brief note to the manuscript when the mutants are first introduced that we are confident wild-type and mutant embryos develop at similar rates and remain comparable until at least early gastrula stage.

Since a novel exciting finding of this study is the fact that opa acts as pioneer factor, it would be exciting to discuss a) the defining properties of a pioneer factor, and b) which of these properties opa seems to fulfil and which ones are still unclear. In such a discussion, a comparison with Zelda, which is the other pioneer factor shown to act as a quantitative temporal timer of gene expression, would be interesting. Additionally, the evidence that opa is able to engage its targets even in the context of nucleosomal DNA is currently absent. I agree that the bioinformatic analysis performed by the authors is consistent with this hypothesis; however, to avoid future confusion, it would be ideal to state it clearly in the Discussion.

We now explicitly compare the observed Opa pioneer activity with that of Zelda in the revised Discussion. Overall, in terms of the fraction of directly-bound sites, Opa and Zld have a similar effect on chromatin accessibility. Schulz et al., 2015 determines that loss of zelda results in ~28% of sites losing accessibility. This is comparable to the ~30% we observe for loss of Opa. We have also framed this within the discussion of what makes a pioneer a pioneer: although we haven’t directly demonstrated binding of Opa to a nucleosome-occluded motif, our measurements are consistent with this possibility, and we point out that additional work needs to be done on this for a definitive answer.

Could the authors provide statistics: how many enhancers are analysed in Figure 3C, D and in Figure 1A? In Figure 3E, the number of peaks is indicated, but as an enhancer can exhibit multiple peaks, it's difficult to infer the number of CRMs considered.

Figure 3C (PCA) is performed on the entire set of peaks with differential expression as determined by DESeq2. This is not selective for enhancers. The number of CRMs in the segmentation network (Figures 1A, 3D) was described in the text accompanying Figure 3. We have added a tabulation of included CRMs in the text accompanying Figure 1. See the response to the first specific comment just below for numbers.

Specific comments:– The manuscript starts by focusing on a small set of known CRM within the segmentation network. Could the authors give a number for the number of segmentation network enhancers for which they observe a dynamic change in chromatin accessibility? Would it be possible to extend this analysis to another type of enhancers, such as the D/V patterning network?

Our list of segmentation network enhancers derived from the Redfly database is 111 sites long. These numbers were in the original submission when we discuss patterning-dependence within this network: 18 gap gene enhancers, 60 pair-rule enhancers, and 33 segment polarity enhancers. For clarity, we have now added to the text accompanying Figure 1 the following additional tabulations: 5/18 (27.8%) gap gene enhancers, 17/60 (28.3%) pair-rule enhancers, and 11/33 (33.3%) segmentation polarity enhancers change accessibility over time. Overall 29.8% of enhancers in this network undergo a statistically-significant change in accessibility over time.

We have not extended this analysis to the D/V network because here we chose to use the lateralizing allele of Toll (*RM9*), which retains some amount of Dorsal expression (cited in the text). If Dorsal influences accessibility patterns, this activity may remain intact in these *Tl* mutants, and we would therefore underestimate the contribution of Dorsal to accessibility. In a future study, we will be modulating D/V fates to address this question directly. To this end, however, we included in the original submission a section in the Discussion dedicated to the question of whether accessibility needs to be modulated along the D/V axis by maternal factors. We predict that it does not, at least early on. Patterning along the A/P axis is determined by single genes with multiple stripes at all levels of network organization, and we observed (Hannon, 2017) that one function of Bicoid is to pioneer anterior-domain specific enhancers of *knirps* and *giant*, as well as *eve1* and a slew of the “head gap genes” amongst others. Early D/V genes, on the other hand activate in single domains along the D/V axis and may therefore not require conditional accessibility states to generate sufficient spatial expression pattern complexity. The definitive test will be to compare accessibility in an allelic series of *Tl* mutants. It remains possible, however, that changes in accessibility accompany cell lineage commitment and distinguish, for instance, mesoderm, neurectoderm, and ectodermal trajectories. In support of these observations, the single-cell ATAC study of record (Cusanovich) identified several distinct early clusters of cells distinguished along the A/P axis, but the only D/V-distinct clusters arose in more advanced stages and delineated mesodermal and neurectodermal signatures.

To extend our observations, we have now compared our list of differentially-accessible regions with the Kvon/Stark set of experimentally validated enhancer elements (Vienna Tiles). These results estimate which additional systems may show high overrepresentation within our different groupings of dynamic regions. As discussed in the response below, we have tried to carefully balance the information overload of a genomics study with extracting meaningful information about embryonic development. Hence, we have maintained the central focus on the segmentation network, but have expanded the scope of the analysis to lend support to the idea that this phenomenon isn’t necessarily just limited to this network.

– Since the ATAC-seq data have been performed in carefully staged single embryos, it would be interesting to extract more information than the small set of segmentation enhancers.

We chose to focus on the segmentation enhancers because they, for the most part, represent a comprehensive GRN that would be definitively disrupted with our genetic approach. By focusing on a comprehensive network, we feel that this focuses the genomic analysis back on to the embryo and away from more abstract heat maps, gene sets, and p-values. We have provided the raw and processed data, as well as comprehensively annotated peaks lists (with outputs of differential enrichment tests) for other investigators to query, and certainly we have not exhausted the analytic possibilities for this dataset.

One less-well characterized presumptive GRN would be the one that drives early posterior endodermal development in the fly. In our experiments, we have identified likely a set of putative CRMs that operate within this lineage in the group of peaks that have enhanced accessibility in the mutant compared with wild-type. We now highlight these regions in the Results section associated with Figure 4, and we explicitly discuss this group in the Discussion.

– Since the authors performed opa ChIP-seq, can they examine if there is a correlation between the number of opa binding sites and the timing of accessibility?

We have addressed this in the following way: we counted the number of Opa motifs (at 80% match to the PWM reported by the MEME analysis associated with the Opa ChIP peaks) over the set of ChIP peaks. Overall, an Opa ChIP peak contains between 0 and 9 Opa motifs (mean = 1.08). The set of direct Opa targets that require Opa for accessibility have between 0 and 9 motifs (mean = 1.32). In contrast, the set of direct Opa targets that do not require Opa for chromatin accessibility have between 0 and 5 motifs (mean = 0.95). This indicates that Opa-dependent regions have on average a

higher number of Opa motifs than Opa-independent regions, although the magnitude of this difference strikes us as small. The difference between these values is statistically significant by both a Wilcoxon Rank Sum test (*p =* 4.026e-6), as well as a one-tailed permutation test on the difference between the mean Opa motif number in proportional randomly selected groups (1e+7 trials, *p =* 2e-7). We have reported this distribution of motifs across the Opa-dependent and -independent groups, the average number of motifs between groups, and the p-value of the permutation test. We have also appended a column to both the Opa ChIP peaks, as well as the ATAC peaks reporting the number of Opa motifs within each peak region.

– Could the authors discuss similarities and differences with the pioneer factor Zelda, as several studies (Foo et al., 2014, Dufourt et al., 2018, Yamada et al., 2019) demonstrate its function as a quantitative timer?

We have done this in the revised Discussion.

– In Figure 6, the authors employ a gene expression analysis on a subset of opa-dependent CRM to support the conclusion that 'the primary function of opa is to modulate the temporally restricted accessibility of a subset of critical CRM'. Although not essential for this manuscript, it would be very exciting to build MS2 reporter transgenes for these critical CRM (for example odd-late) and examine the effect of adding extra opa sites to see if this is sufficient to elicit a premature expression (again like Zelda does, as shown in Yamada et al. and Dufourt et al.).

We agree that this would be an interesting analysis and thank the reviewer for the suggestion. We have gotten as far as co-imaging Opa protein expression and MS2 reporters of late pair-rule loci such as *eve-late* and *odd-late*. We have not yet begun the intensive testing of several mechanistic hypotheses related to enhancer architecture and Opa function such as the one suggested above. This overall represents a significant body of work that will be addressed in a future study.